# Neuropilin-2/Semaphorin-3F-mediated repulsion promotes inner hair cell innervation by spiral ganglion neurons

Thomas M Coate[1,2]*, Nathalie A Spita[2], Kaidi D Zhang[2], Kevin T Isgrig[1], Matthew W Kelley[1]

[1]Laboratory of Cochlear Development, National Institute on Deafness and Other Communication Disorders, Bethesda, United States; [2]Department of Biology, Georgetown University, Washington, United States

**Abstract** Auditory function is dependent on the formation of specific innervation patterns between mechanosensory hair cells (HCs) and afferent spiral ganglion neurons (SGNs). In particular, type I SGNs must precisely connect with inner HCs (IHCs) while avoiding connections with nearby outer HCs (OHCs). The factors that mediate these patterning events are largely unknown. Using sparse-labeling and time-lapse imaging, we visualized for the first time the behaviors of developing SGNs including active retraction of processes from OHCs, suggesting that some type I SGNs contact OHCs before forming synapses with IHCs. In addition, we demonstrate that expression of Semaphorin-3F in the OHC region inhibits type I SGN process extension by activating Neuropilin-2 receptors expressed on SGNs. These results suggest a model in which cochlear innervation patterns by type I SGNs are determined, at least in part, through a Semaphorin-3F-mediated inhibitory signal that impedes processes from extending beyond the IHC region.

*For correspondence: tmc91@georgetown.edu

**Competing interests:** The authors declare that no competing interests exist.

**Reviewing editor**: Graeme W Davis, University of California, San Francisco, United States

## Introduction

The perception of sound in mammals depends on appropriate connectivity between spiral ganglion neurons (SGNs) and hair cells (HCs) in the cochlea. SGNs are bipolar afferents linking the peripheral and central auditory systems by projecting peripheral axons that form ribbon-type synapses with cochlear HCs, and central axons that synapse with neurons in the ipsilateral cochlear nucleus (*Nayagam et al., 2011*). It is known that SGN loss is a key aspect of congenital or noise-induced hearing loss, but only a small handful of mechanisms controlling the formation of this part of the auditory system have been identified (*Fritzsch et al., 2004*; *Appler and Goodrich, 2011*; *Green et al., 2012*; *Coate and Kelley, 2013*). During mouse development, otic neuroblasts delaminate from the otocyst around embryonic day 8.5 and then differentiate into either vestibular or auditory (spiral ganglion [SG]) neurons (*Appler and Goodrich, 2011*; *Coate and Kelley, 2013*). Subsequently, the SGN somata extend in a spiral that mirrors the adjacent developing cochlear duct (~E11.5-P0) while projecting peripheral processes (sometimes termed 'peripheral axons' or 'dendrites') into the sensory epithelium and central axons into the brainstem. The dependence of neurotrophins for the survival and viability of SGNs during these stages has been documented extensively (*Fritzsch et al., 2004*; *Green et al., 2012*), but relatively little is known about the mechanisms controlling SGN diversification and wiring with specific cochlear HC regions during development.

Although SGNs show little morphological variability, they are clearly diverse in several ways. First, SGNs are divided into two broad categories: type I and type II. Type I SGNs represent 90–95% of the total SGN population, form monosynaptic contacts with individual inner HCs (IHCs), and facilitate the majority of all hearing function. Type II SGNs comprise the remaining 5–10% of the SGN population, and, in contrast with Type I SGNs, bypass the IHCs to form *en passant* synapses with up to 10 outer HCs

**eLife digest** The process of hearing begins when sound waves enter the outer ear, causing the eardrum to vibrate. The three small bones of the middle ear pass these vibrations on to the cochlea, a fluid-filled structure shaped like a spiral. Tiny hair cells inside the cochlea move in response to the vibrations and convert them into electrical signals, which are transmitted by cells called spiral ganglion neurons (SGNs) to the brain.

Hair cells can be divided into 'inner' and 'outer' hair cells. Inner hair cells transmit most of the information about a sound to the brain, via connections with type I SGNs. Outer hair cells are thought to amplify sound and connect to type II SGNs. How the type I and II SGNs connect to the correct type of hair cell as the ear develops is not well understood, despite these connections being essential for hearing.

Coate et al. have now used time-lapse imaging and fixed specimens to follow individually labeled SGNs as they establish these connections within the cochlea of a mouse embryo. Although the type I SGNs ultimately formed connections with inner hair cells, many of them made contact with outer hair cells first. These contacts were short-lived thanks to a protein found near the outer hair cells, named Semaphorin-3F. This protein repels the type I SGNs by activating a receptor on their surface called Neuropilin-2, and so directs the type I SGNs towards the inner hair cells.

One of the mysteries that remains to be solved is how type II SGNs are 'permitted' to extend into the outer hair cell region, even though they are also confronted by Semaphorin-3F. In addition, it will also be important to determine how SGNs adapt to cues from different Semaphorins from different parts of the cochlea as they navigate into different hair cell regions.

(OHCs) per individual SGN fiber (*Figure 1A*). In addition, type II SGNs are not myelinated by the neural crest derived-Schwann cells that myelinate type I SGNs (*Carney and Silver, 1983*; *Breuskin et al., 2010*). Although their function is not well understood, recent landmark studies have suggested that type II SGNs may facilitate responses to very loud or painful sounds (*Weisz et al., 2009*, *2012*, *2014*). SGNs also show differences in protein expression and firing rates that correlate with their location along the tonotopic axis (*Flores-Otero et al., 2007*). Finally, type I fibers can be sub-divided into two groups based on spontaneous firing rates and synaptic location at the base of individual IHCs (*Liberman, 1982*).

How the innervation patterns for type I and II SGNs are established is a fundamental question in auditory neuroscience that has yet to be completely answered. One of the most basic issues has been the question of whether individual SGN fibers are specified as either type I or type II prior to the arrival of their peripheral processes in the cochlear duct or if phenotype is established based on interactions within the target environment. Echteler used horse radish peroxidase staining to provide evidence that, in the postnatal gerbil cochlea, immature SGNs show unbiased innervation to both HC regions and are subsequently retained in either the IHC or OHC region likely through a process of selective pruning and apoptosis (*Echteler, 1992*; *Echteler et al., 2005*). Similarly, rhodamine-dextran dye-labeling of type I SGNs in mouse suggested that HC innervation was non-specific until about postnatal day 3 when refinement events seemed to commence (*Huang et al., 2007*). However, genetic-labeling experiments in which sparse numbers of SGNs expressed alkaline phosphatase under the control of Cre recombinase (driven by *Neurogenin-1* promoter elements; *Neurog1^CreERT2^*) suggested that 'mature' SGN innervation patterns are present by birth (*Koundakjian et al., 2007*). These results provide conflicting conclusions regarding the timing and determination of type I and type II SGNs. We sought to gain clarity on this issue by combining the same *Neurog1^CreERT2^* allele with a tdTomato reporter in order to characterize the timeline of IHC and OHC innervation by the SGNs. In addition, we designed experiments to investigate potential molecular mechanisms controlling the differential innervation of IHCs and OHCs and discovered a prominent chemorepulsive role for Neuropilin-2 and its ligand, Semaphorin-3F.

## Results

### Development of type I and type II SGN morphologies occurs prior to birth in mice

Because existing data were somewhat unclear regarding the developmental timing of type I and type II SGN projection patterns, SGN processes and nascent HCs were visualized in cochlear cross sections

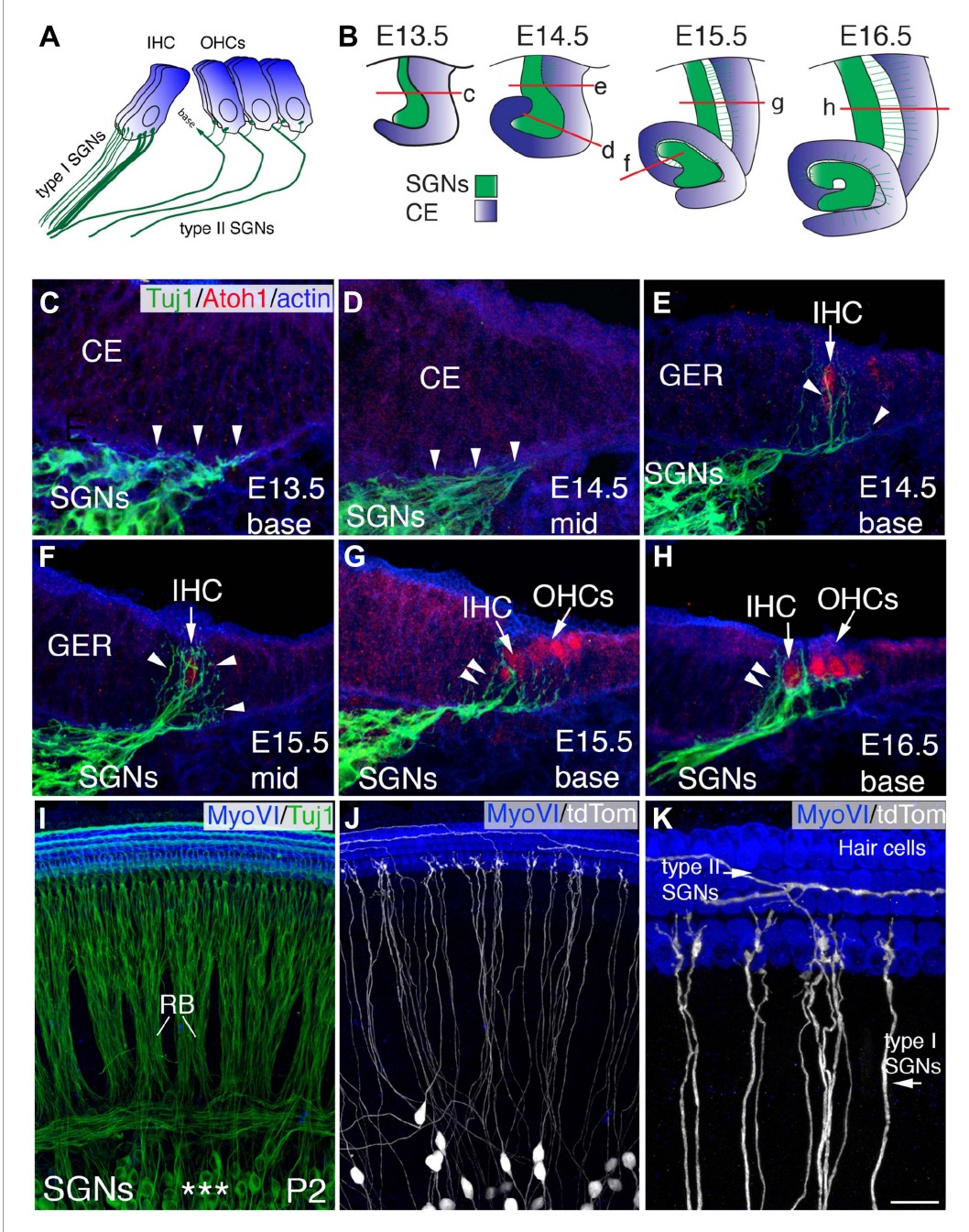

**Figure 1**. Development of SGN innervation patterns. (**A**) Illustration of the innervation pattern of inner hair cells (IHCs) and outer hair cells (OHCs) by type I and II spiral ganglion neurons (SGNs). (**B**) Illustrations showing the morphology of the cochlear duct and spiral ganglion (SG) for the indicated cross-sections. Letters on each illustration correspond to the position of the section in the adjacent panels. ce, cochlear epithelium. (**C–H**) Cross-sections of the cochlea and associated SG at the indicated time points and positions. SGN processes are labeled with Tuj1 (green), hair cells (HCs) with anti-Atoh1 (red), and actin with phalloidin (blue). SGN processes do not enter the epithelium until HCs being to differentiate (E14.5 base) but then rapidly extend towards the developing IHCs (**C–E**). As OHCs begin to develop at E15.5, some processes extend past the IHCs to form contacts with OHCs (**F–H**). (**I**) Confocal image of flat-mounted cochleae from a *Neurog1^{CreERT2}; R26R-tdTomato* mouse at P2. HCs are labeled with anti-myosin VI (blue) and all SGNs are labeled with Tuj1 (green). Asterisks mark the SGN somata, which extend SGN fibers that form into radial bundles (RB) prior to forming synapses with HCs. (**J**) The same preparation as in panel **I**, but illustrating expression of tdTomato (anti-dsRed) to visualize sparsely labeled individual fibers. *Figure 1. continued on next page*

*Figure 1. Continued*

(**K**) High-magnification view from panel **J** illustrating type I SGNs innervating IHCs and type II SGNs passing IHCs to innervate OHCs. Scale bar in **K**: 20 µm, **C**–**H**; 45 µm, **I** and **J**; 15 µm, **K**.

from different developmental time points (*Driver et al., 2013*) (*Figure 1B–H*). At embryonic day 13.5 (E13.5), prior to detectable expression of Atoh1, SGN processes are present outside of the duct, but do not yet project into the cochlear epithelium (*Figure 1C*). By one day later (E14.5), processes have entered the epithelium at the more mature base of the cochlea while remaining outside in less mature, more mid-modiolar regions (*Figure 1D,E*). Neurite entry correlates with the onset of Atoh1 expression in developing HCs (*Figure 1E*; arrow). At E15.5, the ratio of SGN processes projecting towards the emerging OHC region appears to be closer to 50:50 than to the mature ratio of 5:95 (*Figure 1F,G*; see arrowheads). However, at E16.5, a noticeably higher concentration of processes appears around the IHC region (*Figure 1H*; see arrowheads). These data suggest that the SGNs may assume a type I- or type II-like morphology as early as E16.5, which is both consistent with the time period of hair and supporting cell maturation and earlier than previously suggested. Consequently, we focused on these earlier time points in subsequent investigations into the cellular and molecular nature of IHC and OHC innervation.

Approximately, 10–15 individual SGN peripheral axons project to each IHC, making analysis of changes in individual fibers difficult (*Figure 1I*). To circumvent this problem, mice carrying an inducible *Neurog1*$^{CreERT2}$ BAC transgene (*Koundakjian et al., 2007*) were crossed with an *R26R*$^{tdTomato}$ reporter line (referred to as N$^{Cre}$R$^{Tom}$). By limiting recombination to a small number of SGNs, individual fibers with type I- and type II-like morphologies can be clearly visualized (*Figure 1J,K*). The intensity of the tdTomato fluorophore also permitted the resolution of small exploratory protrusions on individual SGN processes (*Figure 1K*).

## Maturing SGNs extend and then retract projections from the OHC region

To better understand the temporal and spatial dynamics of SGN process development, we visualized sparsely labeled individual SGNs (see 'Materials and methods' for details) in whole-mount preparations of cochleae from N$^{Cre}$R$^{Tom}$ mice at different developmental time points (*Figure 2*). The mice also carried the *Atoh1*$^{nGFP}$ transgene (*Lumpkin et al., 2003*), which allows for easy identification of IHCs and OHCs. At the mid-point along the cochlea at E15.5, approximately 60% of individual SGN fibers terminated in the IHC region while the remaining 40% extended into the OHC region (*Figure 2A,B,E*). However, in a more basal region of the same cochlea, the number of fibers terminating in the IHC and OHC regions was roughly equivalent (*Figure 2E*). These results were comparable to the distribution of total fibers that was observed in cross-sections (*Figure 1F,G*). In contrast, at E17.5 or 18.5, between 70 to 80% of all labeled processes were apposed to IHCs (*Figure 2C–E*), suggesting that a progressive transition to the mature innervation pattern begins shortly after E15.5. Consistent with this trend, at P0 and through P3, approximately 90–95% of all labeled SGNs fibers terminated in the IHC region with the remaining 5–10% extending into the OHC region (*Figure 2E*; see also *Figure 1J,K*). Previously, studies using the postnatal gerbil showed that programmed cell death may contribute to the final distribution of type I and II SGNs (*Echteler et al., 2005*). To determine if apoptosis could play a role in SGN innervation patterns during embryonic mouse development, we examined cleaved caspase-3 (CC3) immunostaining at E16.5 and E17.5 (*Figure 2—figure supplement 1*; see 'Materials and methods'). In the basal half of the cochlea at these stages, where most of our neuronal-tracing experiments were performed, we detected between 4 and 19 CC3-positive SGNs per cochlea. At the apex, between 18 and 50 CC3-positive SGNs were observed per cochlea. Given that the number of SGNs is estimated to be greater than twenty thousand at this stage (*Richter et al., 2011*), a role for apoptotic cell death in SGN innervation at this stage seems unlikely.

These data are consistent with a model in which a percentage of SGN fibers initially extend into the OHC region before retracting to the IHC region starting at approximately E16.5. However, an alternate, or possibly, additional hypothesis would be that the population of SGNs observed at E15.5 is largely stable and that the change in the ratio of axon distribution is a result of the arrival of new

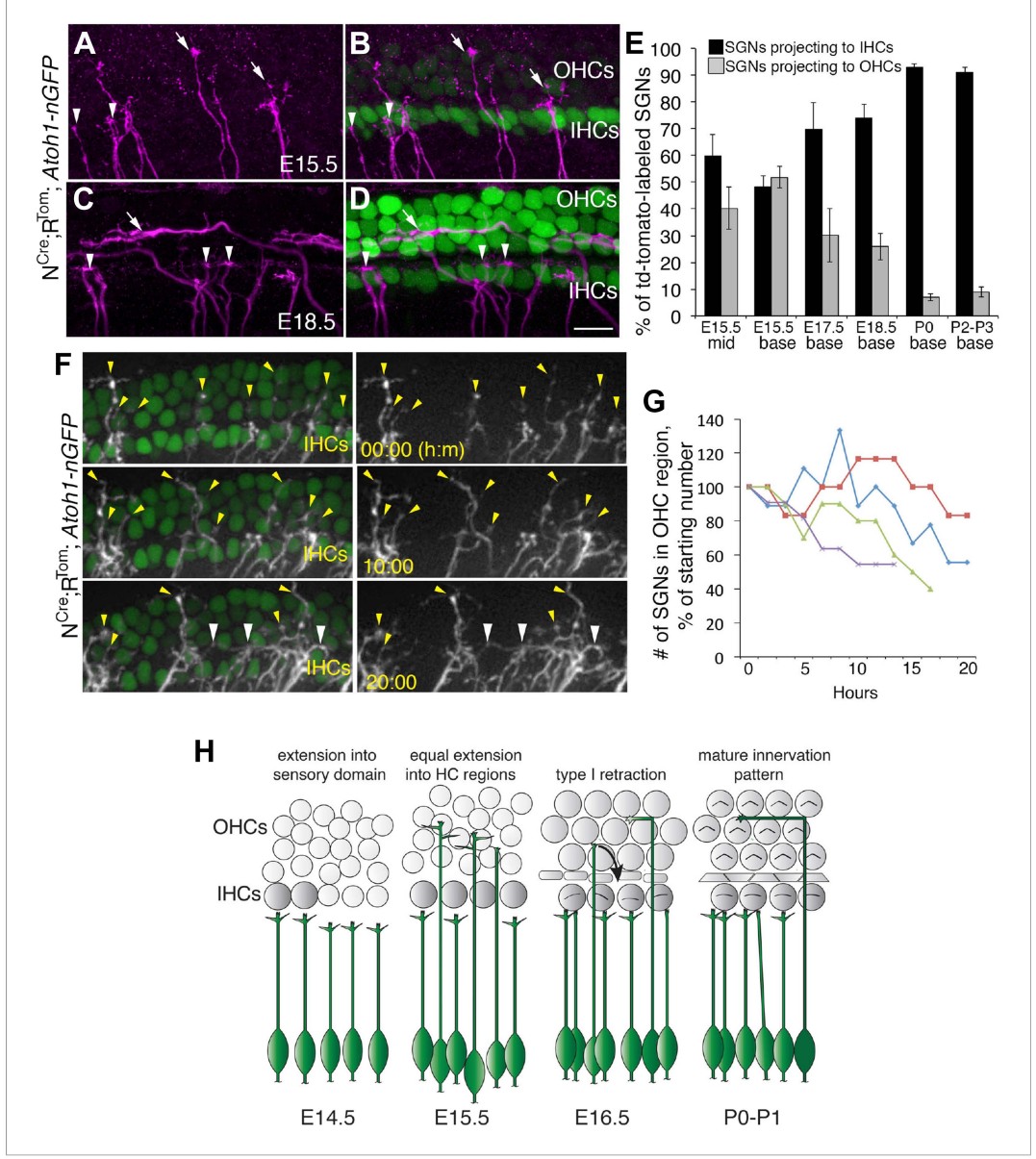

**Figure 2**. Sparse SGN labeling reveals retraction of processes from the OHC region during embryonic development. (**A**–**D**) Fixed whole-mount cochleae from *Neurog1^CreERT2*;*R26R^tdTomato*; *Atoh1^nGFP* mice at E15.5 (**A**, **B**) and E18.5 (**C**, **D**). Panels **A** and **C** show sparse-labeling of SGN fibers (tdTomato in magenta), while panels **B** and **D** show the same regions but include HCs in green. Note that at E15.5, the number of projections terminating in the IHC (arrowheads) and OHC (arrows) regions appears roughly equivalent. In contrast, at E18.5, the majority of SGNs terminate in the IHC region. (**E**) Comparison of the percentage of labeled SGNs that terminate in the IHC (black bars) or OHC region (gray bars) at different developmental time points (minimum of 6 cochleae per time point). Error bars are standard error of the mean (sem). (**F**) Sequential images at the indicated times from a single time-lapse experiment showing sparsely labeled SGNs with HCs (left) or alone (right) in an E16.5 cochlear explant. Yellow arrowheads indicate SGN fiber terminals in the OHC region. White arrowheads point to SGNs that are clustered around IHCs that had originally been in the OHC region. (**G**) Data from four imaging experiments illustrate a decrease in the number of fibers positioned in the OHC region over the course of experimental time. Since there was variability in the number of labeled SGNs between experiments, the data for each experiment were normalized to the number of labeled fibers present in the OHC region at time 0. (**H**) Illustration of the time-line of cochlear innervation. Fibers initially arrive at the IHC region around E14.5. At E15.5, when both IHCs and OHCs are present, SGN fiber distribution is roughly equal between the IHC and OHC regions. In contrast, by E16.5, many SGN fibers have withdrawn from the

*Figure 2. continued on next page*

*Figure 2. Continued*

OHC region. At P0, the vast majority of SGNs terminate at the IHC region. These events occur before the final stages of synapse formation and subsequent fiber pruning. Scale bar in **D**: 15 µm, **A–D**; 20 µm, **F**.

The following figure supplement is available for figure 2:

**Figure supplement 1**. Programmed cell death (apoptosis) is limited in SGNs at E16.5 and E17.5.

SGN peripheral fibers in the IHC region. These new fibers would effectively decrease the percentage of fibers projecting to the OHC region. To determine whether existing OHC fibers retract during development, we performed time-lapse imaging using cochleae from N$^{Cre}$R$^{Tom}$;*Atoh1$^{nGFP}$* mice for up to 20 hr starting at ages E15.5 or E16.5 (see 'Materials and methods'). Analysis of time-lapse videos indicated numerous instances of SGN processes that extended into the OHC region and then retracted back into the IHC region or that were present in the OHC region at the beginning of the experiment and then retracted (*Figure 2F*; *Video 1*). We quantified this effect for axons in four independent time-lapse experiments and found that 10–60% of the axons that started in the OHC region had retracted by the end of the acquisition period (*Figure 2G*). Because of the thickness of the tissue, we were not able to resolve the final location of each process that retracted from the OHC region; however, many of them appeared to aggregate around the IHCs (white arrowheads; *Figure 2F*). In addition, although we cannot rule out that it occurs at all, we never identified new SGNs approaching the IHCs in these experiments. Overall, these data indicate that many SGN peripheral axons initially extend into the OHC region but subsequently retract to form connections with IHCs. Moreover, these results suggest that most SGNs have determined their phenotype (type I vs type II) by the late embryonic period or just shortly after birth (*Figure 2H*).

## Nrp2 is expressed in SGNs and is required for a normal pattern of innervation

The results described above suggest that development of SGN innervation patterns may be influenced by environmental cues within the developing organ of Corti. The Neuropilins are a well-known class of axon-guidance receptors that are activated by secreted Semaphorins (Sema3s) and play diverse roles in innervation of both the central and peripheral nervous systems (*Kolodkin and Tessier-Lavigne, 2011*). Moreover, our previous unpublished studies using RT-PCR demonstrated that *Neuropilin-2* (*Nrp2*) is expressed in whole cochleae at E16.5. Therefore, to determine the temporal and spatial distribution of Nrp2, we localized Nrp2 protein (*Giger et al., 2000*) on cross-sections of developing cochleae and counterstained with Tuj1 to mark SGN somata and their processes (*Figure 3A–H*). For the cross-sections shown here, identical fixation, immunostaining, and imaging parameters were used throughout. At E14.5, Nrp2 was detectable in the SGNs, but not in surrounding mesenchyme or epithelial tissues (*Figure 3A,B*). At E16.5, when SGNs develop region-specific innervation patterns, Nrp2 is clearly detectable in the somata and peripheral axons of the SGNs (*Figure 3C,D*). The Nrp2 antibody labeled all of the SGN cell bodies, which suggests Nrp2 is expressed by all SGNs. Consistent with this conclusion, Nrp2 was visible along all putative type I and II SGN endings in cross-sections at E18.5 (*Figure 3E,F*) and in whole-mount preparations at P0 (*Figure 3G,H*). Interestingly, while cross-sections from P0 and P5 cochleae indicated persistent expression of Nrp2 protein in SGN

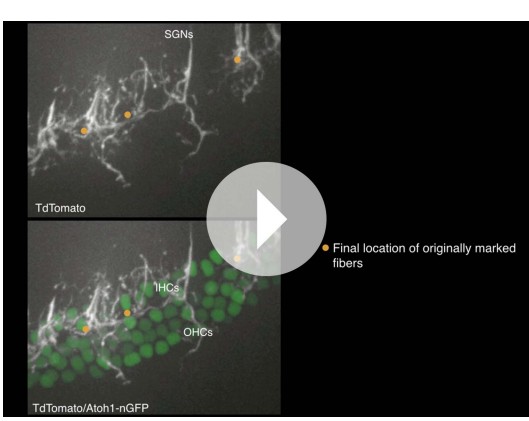

**Video 1.** A cochlea from a mouse carrying the *Neurog1$^{CreERT2}$*; Rosa-tdTomato; Atoh1-nGFP alleles that was imaged by spinning disk confocal microscopy for approximately 20 hr. The orange dots mark SGN processes that can be seen projecting into the OHC region and then falling back to the IHC region.

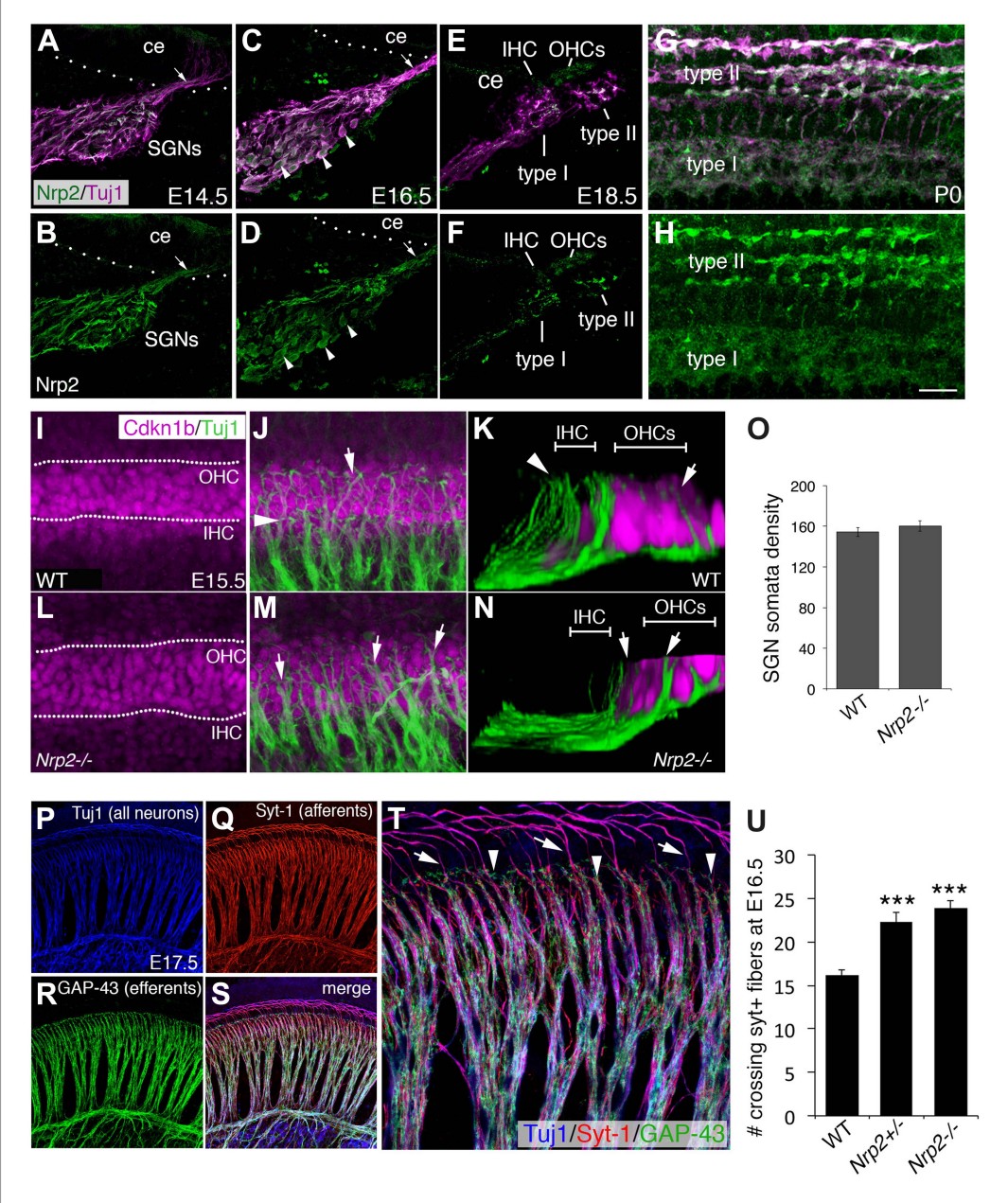

**Figure 3**. Nrp2 is expressed in SGNs and is required for normal HC innervation. (**A–H**) Cross-sections of the cochlear duct and associated SGNs from the indicated ages. Upper panels show both SGNs (Tuj1 in magenta) and anti-Nrp2 (green). Lower panels show anti-Nrp2 alone. (**A, B**) At E14.5, Nrp2 is present in SGN somata and in peripheral axons (arrow) projecting into the cochlear epithelium (ce). (**C, D**) At E16.5, Nrp2 is detectable in virtually all SGN somata (arrowheads). (**E, F**) At E18.5, Nrp2 is detectable in putative type I and type II processes. (**G, H**) Whole-mount preparation at P0 indicates expression of Nrp2 in both type I and type II SGNs. (**I–N**) Comparison of SGN process distribution at E15.5 in WT and *Nrp2⁻ᐟ⁻* cochleae. Anti-Cdkn1b (magenta) labels the developing cochlear prosensory cells prior to HC formation (**I**) while Tuj1 (green) marks the SGNs (**J**). The dotted lines in **I** delineate the IHC and OHC regions. (**K**) Three-dimensional Y-Z view of the processes in **J**. Note that roughly equal number of processes terminate in both the IHC and OHC regions. In contrast, in a cochlea from an *Nrp2⁻ᐟ⁻* mutant, more processes appear to extend into the OHC region (arrowheads in **M, N**). (**O**) Quantification of cell bodies indicates no difference in number of SGNs between WT and *Nrp2⁻ᐟ⁻*. (**P–T**) E17.5 whole-mount cochlea stained with markers for all neurons (Tuj1; blue), SGN afferents (anti-Syt-1; red), and olivocochlear efferents (anti-GAP-43; green). The high-magnification view in **T** is from the same preparation shown in **P–S**. (**U**) Quantification of number of processes crossing into the OHC region in the base of the cochlea at E16.5. Both *Nrp2⁻ᐟ⁻* and *Nrp2⁺ᐟ⁻* mice show significantly higher numbers of

*Figure 3. continued on next page*

*Figure 3. Continued*

SGN processes in the OHC region as compared to WT. ***p ≤ 0.001. n = 9, WT; 6, *Nrp2⁻/⁻*; 4, *Nrp2⁺/⁻*. Error bars, sem. Scale bar in **H**: 35 µm, **A–H**; 20 µm, **F, J, L, M**; 10 µm, **K, N**; 55 µm, **P–S**; 20 µm, **F**.

The following figure supplement is available for figure 3:

**Figure supplement 1**. Semaphorin3F-Fc recognizes binding partners in vivo.

somata, the overall intensity of staining was substantially reduced by comparison with the earlier time points (not shown).

To demonstrate that the Nrp2 immunoreactivity in both SGN subtypes represents functional receptor distribution, we treated cochlear cross-sections with an Fc-tagged version of Semaphorin-3F (a known ligand for Nrp2; *Sahay et al., 2003*) at P0, a stage when type I and II SGNs are morphologically distinguishable (as shown in *Figure 2*). The tissue sections were then immunolabeled with anti-Fc antibodies and imaged by confocal microscopy. *Figure 3—figure supplement 1* shows example preparations counterstained with 4′,6-diamidino-2-phenylindole (DAPI) and anti-neurofilament antibodies. In contrast with control Fc, which showed little to no binding (*Figure 3—figure supplement 1A–D*), Sema3F-Fc showed substantial binding to SGNs along the basal-to-apical axis of the cochlea (*Figure 3—figure supplement 1E–H*; see arrows). Similar to the anti-Nrp2 antibody staining in *Figure 3E–H*, Sema3F-Fc was detectable on neurofilament-positive structures that are consistent with the position of peripheral axons of both type I and II SGNs (*Figure 3—figure supplement 1I–P*). These data suggest that functional forms of Nrp2 are present on both type I and II SGNs.

Given that Nrp2 is important in axon guidance and synaptogenesis in multiple systems (*Giger et al., 1998*; *Chen et al., 2000*; *Jongbloets and Pasterkamp, 2014*) and that Nrp2 is expressed by the SGNs, we hypothesized that Nrp2 was necessary for normal cochlear innervation. To test this, we analyzed SGN process extensions in cochleae from *Nrp2⁻/⁻* mutant mice (*Giger et al., 2000*). As expected, in wild type (WT) cochleae at E15.5, SGN processes are nearly evenly distributed between the IHC and OHC regions of the organ of Corti (*Figure 3I–K*). In contrast, in *Nrp2⁻/⁻* cochleae at E15.5, there were large bundles of SGN processes that extended into the OHC region, with far fewer processes terminating near the IHC region (*Figure 3L–N*). These changes were not a result of a decrease in the total number of SGNs, as quantifying SGN somata density indicated no differences between WT and *Nrp2⁻/⁻* cochleae (*Figure 3O*). In order to quantify changes in SGN patterning in *Nrp2⁻/⁻* mutants, it was necessary to discriminate SGNs from associated olivocochlear efferent fibers. Since both are labeled by pan-neuronal markers such as neurofilament or beta-III-tubulin (Tuj1), we screened over a dozen antibodies against known neuronal factors and found that synaptotagmin-1 (Syt-1) antibodies label the SGN afferent axons but do not mark olivocochlear efferents (*Figure 3P–T*). Triple labeling with Tuj1 (all neurons, blue), Syt-1 (SGN afferent axons, red), and GAP-43 (olivocochlear efferents, green; *Simmons et al., 1996*) in an E17.5 cochlea demonstrates the specificity of the Syt-1 antibody for afferent SGN axons. Next, the number of SGN fibers that cross into the OHC region in WT, *Nrp2⁺/⁻*, and *Nrp2⁻/⁻* cochleae was quantified by labeling with Syt-1 antibodies at E16.5. Results indicate significant increases in the number of crossing fibers in both *Nrp2⁺/⁻* and *Nrp2⁻/⁻* cochleae (*Figure 3U*). These data are consistent with a model in which Nrp2 plays a role in limiting the number of SGN axons that cross into the OHC region.

## *Nrp2* modulates hair-cell targeting and SGN axonal complexity

A somewhat unexpected result of the analysis of Syt1+ SGNs was the observation of similar patterning defects in cochleae from *Nrp2⁺/⁻* and *Nrp2⁻/⁻* animals. However, phenotypic defects in midline commissural axon path finding have been reported in *Nrp2⁺/⁻* mice, suggesting that heterozygosity for *Nrp2* can lead to a significant decrease in Nrp2 protein (*Chen et al., 2000*). To determine if this is the case in SGNs, we examined protein levels by Western blot and found a significant decrease in Nrp in *Nrp2⁺/⁻* cochleae (*Figure 4A*). It was surprising that the *Nrp2⁺/⁻* heterozygous cochleae consistently had so little Nrp2 protein, but previous reports have shown positive feedback/autocrine loops associated with *Nrp2* expression (*Tyler Hillman et al., 2011*; *Goel et al., 2013*). Based on these results and the fact that *Nrp2⁻/⁻* mice demonstrate limited survival past

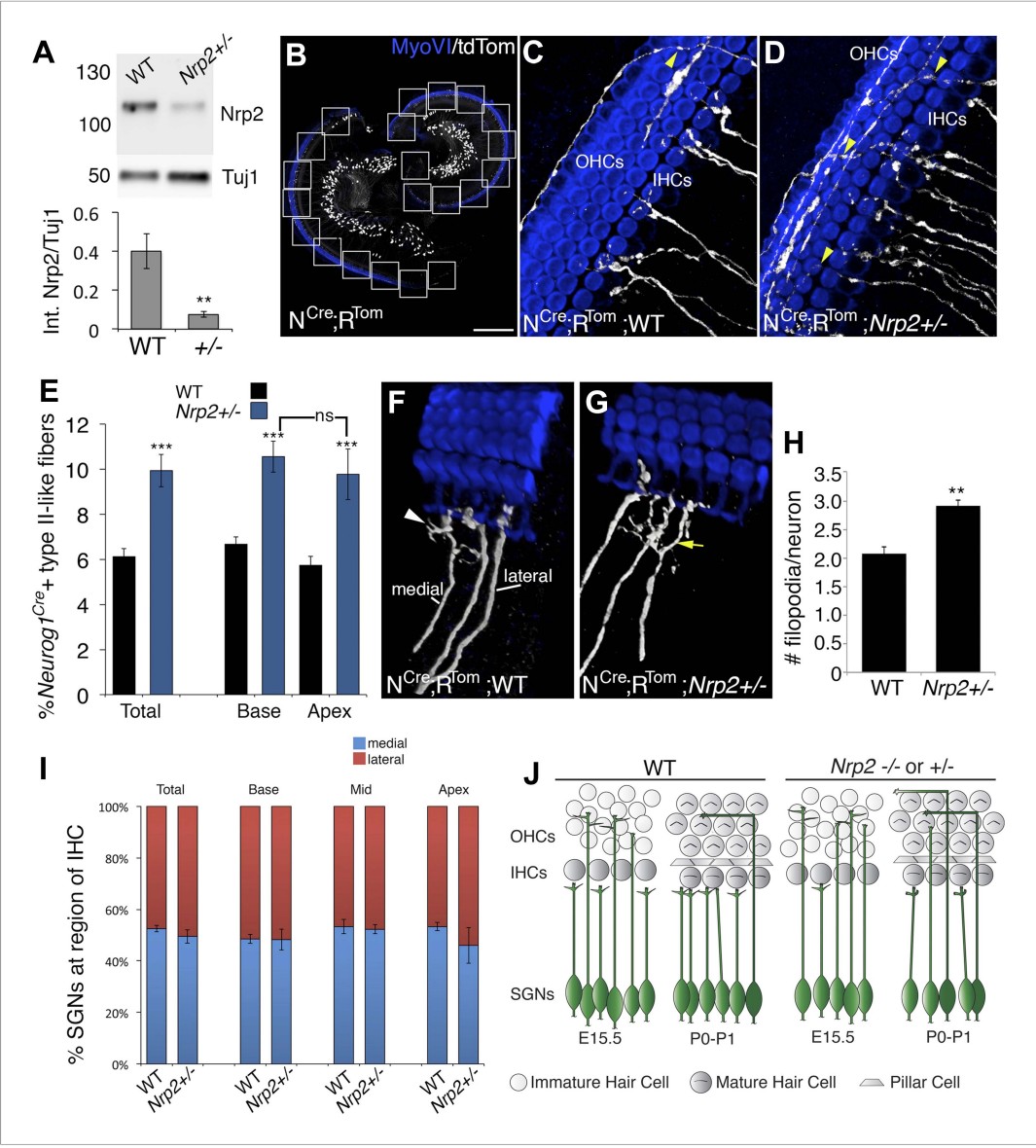

**Figure 4.** Nrp2 regulates SGN axonal complexity. (**A**) Western blot demonstrating significantly reduced Nrp2 protein in cochleae from *Nrp2*[+/−] mice as compared to WT. **p ≤ 0.01; n = 9 WT; 7 *Nrp2*[+/−]. (**B**) Low-magnification view of a *Neurog1*[CreERT2]; *R26R-tdTomato* cochlea with boxed regions showing the locations where confocal z-stacks were acquired. In all of the micrographs in this figure, white represents anti-tdTomato and blue represents anti-myosin VI. (**C**, **D**) Compared to WT, *Nrp2*[+/−] cochleae with sparse SGN labeling show higher numbers of fibers in the OHC region (yellow arrowheads). (**E**) Quantification of the percentage of sparsely labeled type II-like fibers (defined as SGNs that had crossed into the OHC region and turned toward the base) in each genotype. ***p ≤ 0.001; n = 9 WT; 6 *Nrp2*[+/−]; ns, not significant. (**F**, **G**) 3D reconstructions of type I SGNs (relative positions are indicated in **F**). In *Nrp2*[+/−] cochleae, there were many type I SGNs with increased branching (yellow arrow in **G**), suggesting absence of an inhibitory cue. (**H**) Quantification of number of filopodia per individual type I SGN in WT and *Nrp2*[+/−] cochleae indicates significantly higher numbers of small branches in *Nrp2*[+/−] heterozygotes. **p ≤ 0.01; n = 4 WT; 6 *Nrp2*[+/−]. (**I**) The number of medially and laterally positioned type I SGNs does not change in *Nrp2*[+/−] cochleae. n = 4 cochleae each genotype. (**J**) Model illustrating phenotypic changes in cochlear innervation as a result of a decrease or absence of *Nrp2*. Error bars, sem. Scale bar in **B**: 150 µm, **B**; 20 µm, **C**, **D**, **F**, **G**.

birth, we used the *Nrp2*[+/−] mice to examine changes in SGN innervation at postnatal ages. Moreover, in order to unambiguously visualize potential changes in axonal morphology or path finding, the *Nrp2* mutant strain was bred onto the N[Cre];R[Tom] line. This allowed visualization of a limited number of SGN

axons in animals with significantly decreased Nrp2 (*Figure 4B–D*). As expected, an increased percentage of SGN fibers extended into the OHC region in *Nrp2*[+/−] cochleae (*Figure 4C,D*; see arrowheads). To quantify these changes, we determined percentages of labeled fibers with type I- and type II-like morphologies in cochleae from WT and *Nrp2*[+/−] mice. In previous studies, anti-peripherin immunolabeling has been used to label type II SGNs (*Huang et al., 2007*; *Defourny et al., 2013*), but we and others have found significant peripherin expression in type I SGNs through postnatal day 6. Therefore, type I (terminating at IHCs)- and type II-like (extending into the OHC region and turning towards the base) fibers were classified based on morphology (*Figure 4C,D*). Results indicated nearly a 40% increase in type II-like SGN fibers (among the labeled population) in *Nrp2*[+/−] cochleae at P1 (*Figure 4E*), confirming the phenotype observed in embryonic cochleae from *Nrp2* mutants.

Activation of Neuropilins has been shown to alter growth cone motility (*Chen et al., 1997*), an effect that could account for the SGN phenotype. Therefore, we wanted to examine whether decreased levels of Nrp2 led to increases in either the number or length of small exploratory processes at the ends of SGN peripheral axons. To accomplish this, terminals of individual SGN peripheral axons were examined in cochleae from WT and *Nrp2*[+/−] mutants (with sparse SGN labeling) at P1. The ends of the type II SGNs showed tremendous morphological variability at these stages, but we were not able to detect any substantial differences between *Nrp2*[+/−] and WT. However, we were able to systematically measure and count the number of small branches at the ends of type I SGNs (*Figure 4F,G*). WT and *Nrp2*[+/−] SGNs did not show differences in filopodial length (not shown), but *Nrp2*[+/−] SGNs neurites did show a significant increase in the number of filopodial branches (*Figure 4F–H*). Finally, we examined the synaptic locations of type I SGNs in WT and *Nrp2*[+/−] cochleae, as previous work has suggested that low-spontaneous rate fibers synapse on the medial side of IHCs, while high-spontaneous rate fibers synapse on the lateral sides (*Liberman, 1982*). We tracked over 1500 SGNs between *Nrp2*[+/−] and WT cochleae and found no differences in the percentages of medially or laterally positioned SGNs (*Figure 4I*). These data support the hypothesis that Nrp2 mediates an inhibitory signal within SGN fibers that decreases terminal complexity and prevents fibers from extending processes into the OHC region. However, Nrp2 does not appear to play a role in the patterning of fibers within the IHC region.

## The number of ribbon synapses is normal in adult *Nrp2* mutants

SGNs and HCs connect via specialized ribbon synapses (*Sobkowicz et al., 1982*). Given that decreased Nrp2 results in an increase in the number of SGN processes in the OHC region, we speculated that this should result in an increase in the number of synaptic ribbon bodies in OHCs. To investigate this, we labeled and counted Ribeye-positive puncta at the base of IHCs and OHCs in WT and *Nrp2*[+/−] cochleae at P8 and P21 (*Figure 5A–D*). Surprisingly, synapse numbers for both IHCs and OHCs were unchanged between WT and *Nrp2*[+/−] mutants (*Figure 5E*). For these studies, we were also able to examine both cochleae from one surviving *Nrp2*[−/−] mouse at P21. However, no difference in number of ribbon synapses (as compared to controls) was observed in these cochleae either. Consistent with this result, assessment of hearing sensitivity by auditory brainstem response indicated no significant differences between WT and *Nrp2*[+/−] mice (not shown). To further examine innervation in *Nrp2*[+/−] adult cochleae, we compared numbers of type II SGNs between *Nrp2*[+/−] mutants and WT using sparse SGN labeling. Although there was a small increase in type II projections in *Nrp2*[+/−] cochleae as compared to controls, the difference was not significant. Overall, these results suggest that innervation is normal in adult *Nrp2*[+/−] cochleae. The basis for the change in innervation patterns between embryonic/early postnatal and adult *Nrp2*[+/−] animals is unclear. However, after birth, the cochlea goes through a significant phase of synaptic pruning in conjunction with innervation by olivocochlear efferents (*Huang et al., 2012*). Therefore, it seems likely that this pruning mechanism is independent of Nrp2 and can therefore act to correct any increases in SGN fibers in the OHC region.

## Semaphorin-3F is expressed in the OHC region in a complementary pattern to Nrp2

Nrp2 is thought to be activated by several of the secreted Semaphorins, including Sema3B, Sema3C, Sema3D, and Sema3G (*Sharma et al., 2012*). To determine which of these ligands could be activating Nrp2 on SGNs, we localized *Sema3* transcripts by in situ hybridization. *Sema3b* (not shown) and *Sema3c* showed a nearly identical expression pattern; mRNA was detectable in the SG, in the greater

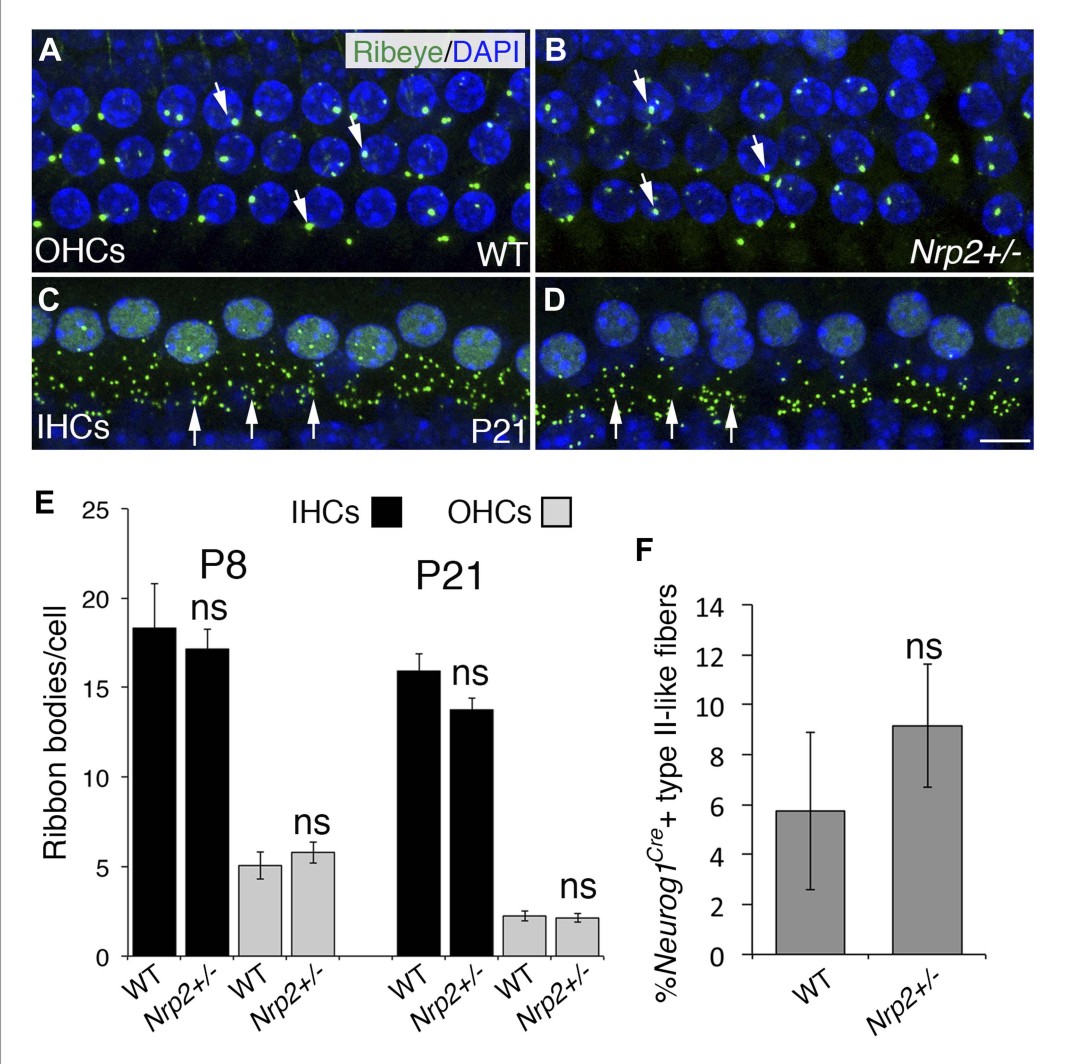

Figure 5. The number of HC ribbon synapses is unchanged in cochleae with reduced *Nrp2*. (**A–D**) Visualization of ribbon synapses (anti-Ribeye in green; white arrows) on IHCs and OHCs at P21. Nuclei are labeled with DAPI (blue). Overall no differences in ribbon synapse numbers were observed between genotypes. (**E**) Quantification of ribbon bodies per IHCs and OHCs in P8 and P21 cochleae. All differences were not significant (ns). n = 6 cochleae per genotype per age. (**F**) Percentage of labeled type II-like fibers (defined as SGNs that had crossed into the OHC region and turned toward the base) in each genotype at P21. n = 6 WT; 6 het. Although the data trended toward more fibers in the *Nrp2*$^{+/-}$ cochleae, the difference compared to the control group was not significant (ns). Error bars, sem. Scale bar, 8 μm.

epithelial ridge (GER; non-sensory cells) of the cochlear epithelium, and very faintly in the IHC and OHCs (*Figure 6A–C*). *Sema3d* mRNA was not detectable in the SG but was detectable in an unusual striping pattern in the GER (*Figure 6D–F*; see the arrow in *Figure 6E*). In situ hybridization for *Sema3g* was not performed, but Richardson et al. did not detect *Sema3g* expression in the E14.5 cochlea (*Richardson et al., 2014*). In contrast with these results, *Sema3f* mRNA is expressed in a pattern highly suggestive of a role in activating Nrp2 (*Figure 6G–N*). At E14.5 (not shown) and E16.5, *Sema3f* transcripts were detectable in the apical, middle, and basal turns of the cochlea in a pattern that is distinct from the other *Sema3s* we examined (*Figure 6H*). Expression of *Sema3f* is present in two regions of the duct: an extreme medial region near the junction with Reissner's membrane and in a lateral domain that correlates with the developing sensory epithelium. At higher magnification, *Sema3f* appears to be expressed in a domain of cells that includes OHCs, pillar cells (PCs), Dieters'

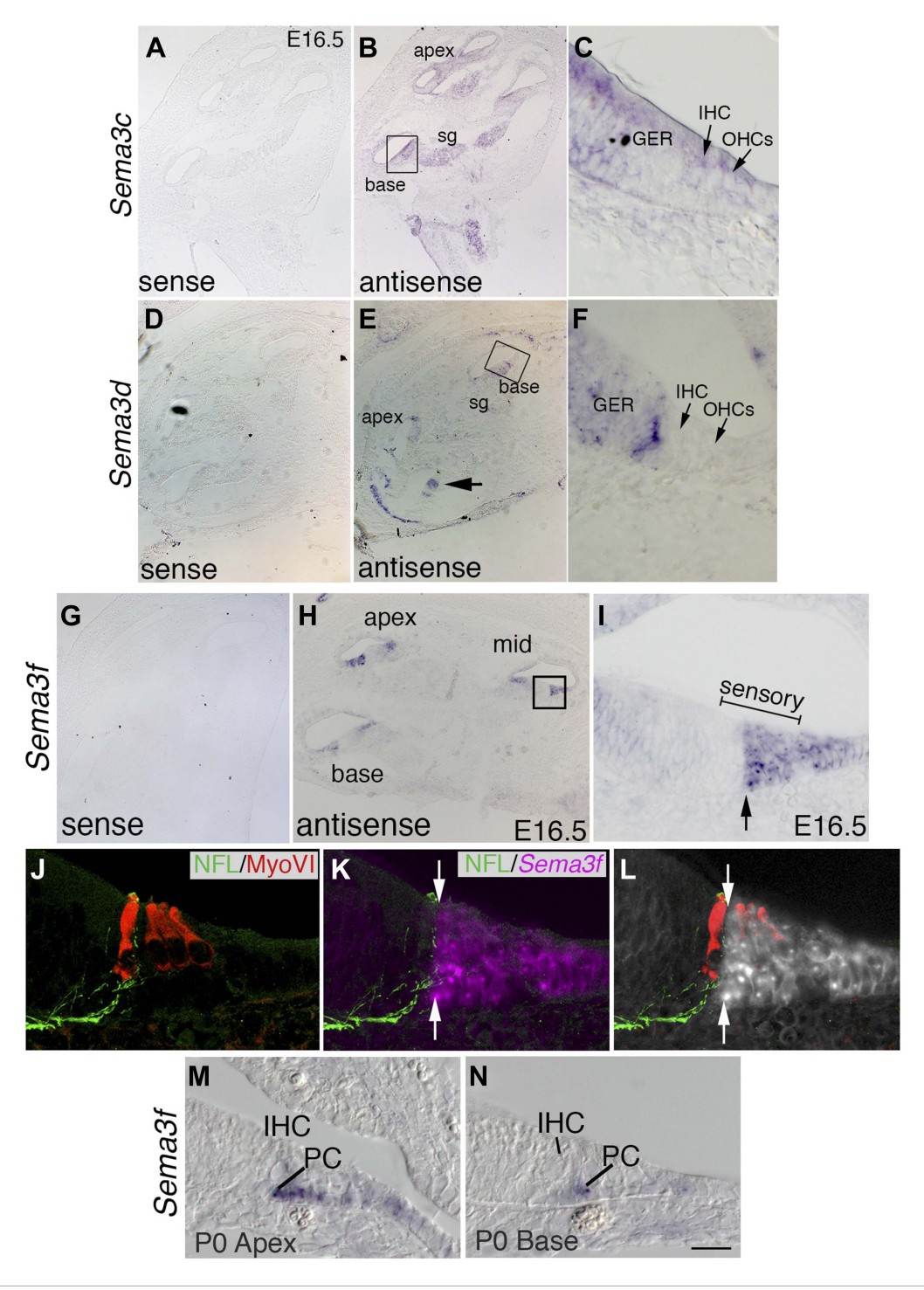

Figure 6. The expression pattern of *Sema3f* suggests a role in Nrp2 signaling during cochlear innervation. In situ hybridizations for *Sema3c*, *Sema3d*, and *Sema3f*. (A–J) Low-magnification sense controls (A, D, G) demonstrate specificity for each probe. Low (B, E, H) and high-magnification views (C, F, J) from the boxed regions in B, E, and H for antisense probes indicate unique patterns of expression. (B, C) *Sema3c* transcripts are present in the SG, greater epithelial ridge (GER), and faintly in the IHCs and OHCs. (E, F) *Sema3d* transcripts are weakly present in the GER and in a striped pattern that includes non-sensory cells within the medial portion of the cochlear epithelium. (H, I) *Sema3f* transcripts are present in the lateral portion of the cochlear sensory domain (bracketed region of I), which includes OHCs, outer pillar cells (PCs), Dieters' cells, and Henson's cells. (J–L) To better resolve the expression pattern for

*Figure 6. continued on next page*

Figure 6. Continued

*Sema3f*, the preparation in panel **I** was counterstained with anti-neurofilament light chain and anti-myosin VI. Results indicate a sharp medial boundary of *Sema3f* expression that correlates with the lateral edge of type I SGN innervation. (**M**, **N**) *Sema3f* expression in the apex and base of the cochlea at P0. IHC, inner HC; PC, pillar cell. *Sema3f* levels are diminished at this stage. By P5, *Sema3f* transcripts are not detectable (not shown). Scale bar in **N**: 350 μm, **A**, **B**, **D**, **E**, **G**, **H**; 20 μm, **C**, **F**, **I**, **M**, **N**; 15 μm, **J–L**.

cells, and Henson's cells (*Figure 6I*). Moreover, the domain has a sharp medial boundary that appears to correlate with the junction between the IHC and OHC regions. To better resolve the spatial distribution of *Sema3f* mRNA, we post-stained samples with antibodies against myosin VI and Neurofilament light chain (*Figure 6J–L*). The merged image demonstrates that *Sema3f* expression clearly overlaps with the OHC region and is completely absent from the IHC region. In addition, the sharp medial boundary appears to precisely correlate with the limiting extension of developing SGN processes. At postnatal day 0 (P0), *Sema3f* expression in the cochlea is diminished and the distribution pattern has changed. At the apex, *Sema3f* is detectable in the PCs and faintly present in nearby supporting cells, while at the cochlear base, *Sema3f* expression is limited to just the outer PCs. At P5, *Sema3f* was not detectable in the cochlear epithelium (not shown). These data suggest that Sema3F protein is distributed in a spatial and temporal pattern consistent with a role in activating Nrp2 on SGNs.

## Exogenous Sema3F inhibits SGN outgrowth in vitro

To test the hypothesis that Sema3F activates Nrp2 in type I SGNs, which would lead to decreased complexity and inhibition of extension, we established cochlear explant cultures from E15.5 or E16.5 mice and maintained them in media containing control Fc or Sema3F-Fc fusion proteins for 24 hr. At the completion of each experiment, all SGN processes were labeled using Tuj1. To observe the fine details of the SGN endings, it was necessary to prepare 3D reconstructions from high-resolution Z-stacks (*Figure 7A–D*). In controls, type I SGN processes located near the IHCs elaborated complex branching patterns with multiple filopodia per process (*Figure 7A,C*). In contrast, SGN processes from cochleae treated with Sema3F-Fc showed mostly blunt endings with very little branching (*Figure 7B,D*). We quantified this effect by counting the total number of SGN branch tips in the IHC region (*Figure 7E*). For experiments initiated at either E15.5 or E16.5, the addition of Sema3F-Fc led to nearly a 50% decrease in the number of SGN branch tips.

Although there was an obvious effect on the morphology of SGN processes following treatment with Sema3F-Fc, the labeling of all SGNs by Tuj1 prevented a determination of whether this change resulted from a full retraction of some SGNs from the cochlear epithelium or a reduction in the number of branches per individual SGN. Therefore, the Sema3F-Fc experiments were repeated using cochleae from N$^{Cre}$;R$^{Tom}$ mice (*Figure 7F–K*). While exposure to Sema3F-Fc did not decrease the number of processes that reached the IHC region (*Figure 7F–G*), there was an obvious decrease in the number of branched protrusions per process (*Figure 7G,I*). To quantify this change, the number of branches per process was determined for greater than 50 SGNs from each treatment group. Results indicate a significant reduction in the number of branches in the presence of Sema3F-Fc (*Figure 7K*). These data are consistent with the hypothesis that a localized source of Sema3F in the OHC region acts to activate Nrp2 in SGN growth cones leading to decreased complexity and extension.

## Deletion of *Sema3f* leads to increased SGNs in the OHC region

To confirm that Sema3F normally acts to prevent SGN fibers from extending into the OHC domain, the innervation phenotype was examined in *Sema3f*$^{–/–}$ mutant mice (*Walz et al., 2007*). To visualize individual fibers, the sparse-labeling alleles were bred onto the *Sema3f* mutant line (*Figure 8A,B*). As predicted, *Sema3f*$^{–/–}$ cochleae showed an increased percentage of SGN processes in the OHC region. Interestingly, when we quantified the increase in crossing processes in *Sema3f*$^{–/–}$ cochleae (*Figure 8C*), we found that it was comparable to that observed in *Nrp2*$^{+/–}$ heterozygotes (*Figure 4*). Compared to littermate controls, we did not observe any changes in either HC or SGN density in *Sema3f*$^{–/–}$ cochleae

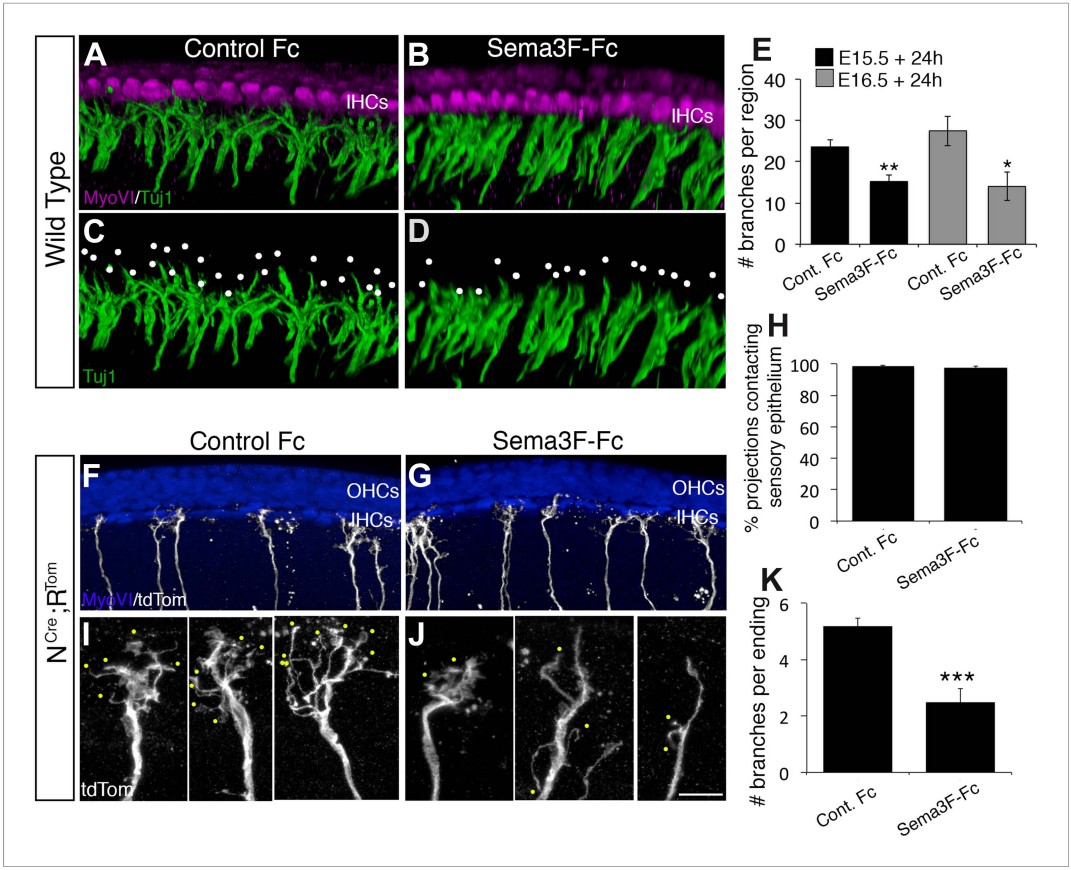

**Figure 7**. Exogenous Sema3f inhibits SGN outgrowth in vitro. (**A–D**) Three-dimensional renderings of confocal z-stacks from cultured cochleae treated with either control IgG-Fc or Sema3F-Fc at 20 nM. The cultures were stained with anti-myosin VI to show the position of the HCs (magenta) and Tuj1 to mark SGNs (green). By comparison with an Fc-treated control (**A**, **C**) in which the SGNs show extensive branching, Sema3F-Fc-treatment significantly inhibited branching (**B**, **D**). The white dots in **C** and **D** mark individual branch points. (**E**) Quantification of SGN branch numbers normalized to the length of the cochlea within the photographed region. For this normalization, we measured the length of the cochlea along the IHC border. Addition of Sema3F-Fc led to significantly fewer SGN branches in explants initiated at E15.5 (black bars) or E16.5 (gray bars). **$p \leq 0.01$; *$p \leq 0.05$. n = 6 cochleae per treatment group. (**F–K**) To determine whether Sema3F-Fc reduced the number of SGNs in the cochlear epithelium or reduced the number of secondary branches per SGN, the Sema3F-Fc experiments were repeated using cochleae from *Neurog1^{CreERT2}*; *R26R-tdTomato* mice. (**F**, **G**) Low-magnification micrographs from cochleae stained with anti-myosin VI (HCs, blue) and tdTomato to mark individual SGNs (white). (**H**) There were no differences in the percentages of labeled SGNs contacting the cochlear sensory epithelium in either group. (**I**, **J**) High-magnification images of representative individually labeled SGN terminals. Yellow dots mark small secondary branches extending from each SGN peripheral process. Note the reduced number of secondary branches in Sema3F-Fc-treated explants. (**K**) Quantification of the number of secondary branches per SGN ending showing the inhibitory effect of Sema3F-Fc. ***$p \leq 0.001$. n = 6 cochleae per treatment group. Error bars, sem. Scale bar in **J**: 15 μm, **A–D**; 30 μm, **F** and **G**; 7 μm, **I** and **J**.

(*Figure 8D–G*), suggesting the extraneous SGN processes in the OHC region are not a secondary effect of other cochlear changes. To determine whether the increase in OHC innervation in *Sema3f^{−/−}* cochleae led to increased ribbon synapse formation, we used the same approach as for Nrp2-deficient cochleae (*Figure 5*). Results indicated no differences in the density of ribbon synapses between control and *Sema3f^{−/−}*. As was suggested for the *Nrp2* animals, these results suggest that synaptic pruning events that occur following innervation probably lead to a correction in the number of SGN fibers present in the OHC region. These data support a model in which Sema3F acts as a ligand for Nrp2 to control SGN extension into the OHC region during development.

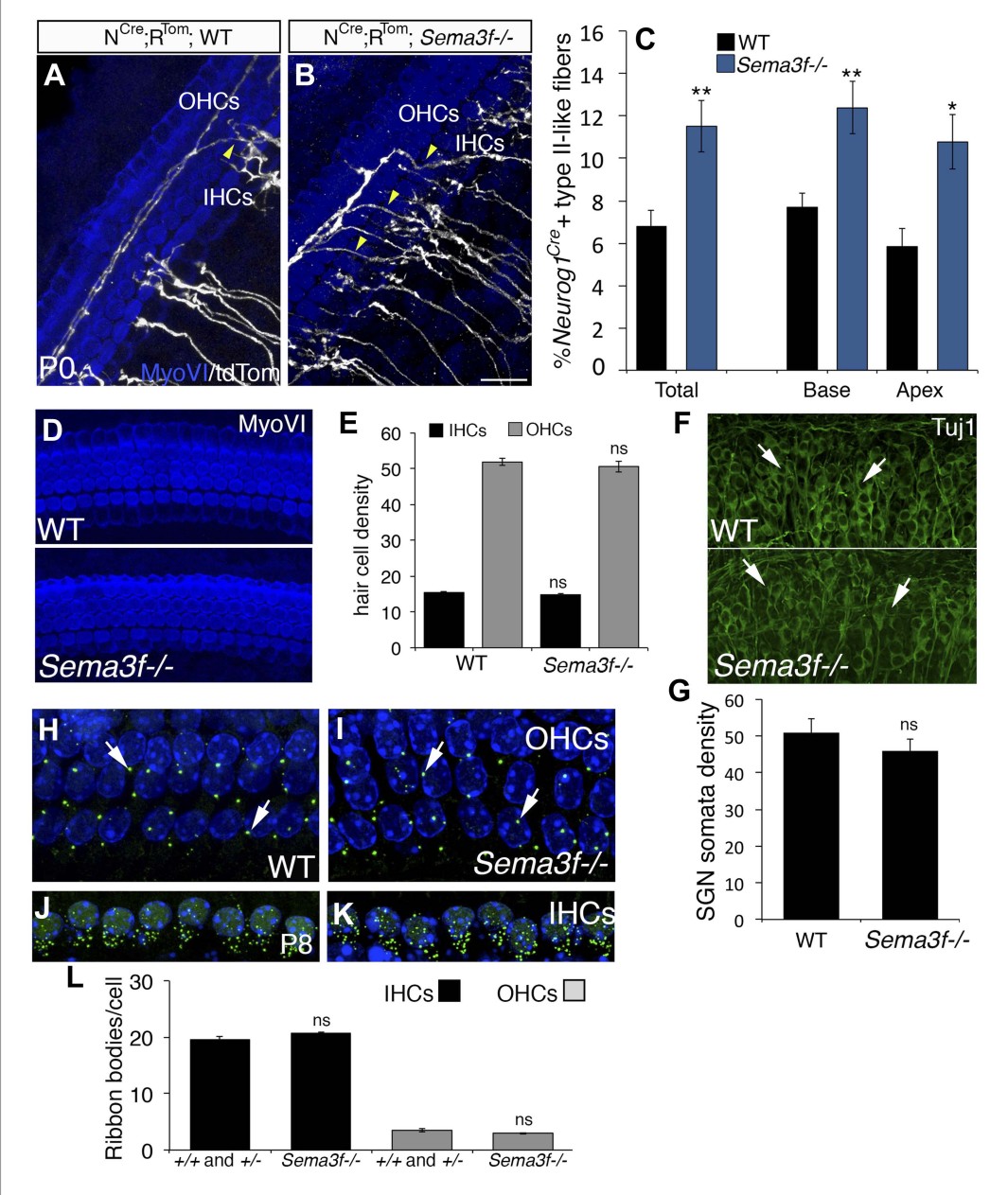

**Figure 8**. Deletion of *Sema3f* leads to increased SGNs in the OHC region. (**A**, **B**) Representative images from *Neurog1CreERT2;R26RtdTom*; WT or *Sema3f−/−* cochleae at P0 stained with anti-myosin VI (HCs, blue) and tdTomato to mark individual SGNs (white). Increased numbers of crossing fibers (yellow arrowheads) projecting into the OHC region are present in the absence of *Sema3f*. IHCs, inner HCs; OHCs, outer HCs. (**C**) Absence of *Sema3f* leads to a significant increase in the percentage of labeled type II-like fibers (defined as SGNs that had crossed into the OHC region and turned toward the base). Note that the change in percentage in the absence of *Sema3f* is comparable to the change observed in *Nrp2+/−* heterozygotes (*Figure 4E*). **p ≤ 0.01; *p ≤ 0.05. n = 6 WT; 6 *Sema3f−/−*; error bars, sem. (**D**) Whole-mount images of HCs in WT and *Sema3f−/−* cochleae after myosin-VI immunostaining. (**E**) Quantification of number of IHCs and OHCs indicates no difference between WT and *Sema3f−/−*. n = 7 each genotype. (**F**) Whole-mount images of SGNs in WT and *Sema3f−/−* cochleae after Tuj1 immunostaining. The white arrows point to individually identifiable SGN somata. (**G**) Quantification indicates no difference in SGN density between WT and *Sema3f−/−*. n = 5 each genotype (**H**–**K**) Visualization of ribbon synapses (anti-Ribeye in green; white arrows) on IHCs and OHCs at P8. Nuclei are labeled with DAPI (blue). Overall, no differences in ribbon synapse numbers were observed between genotypes.
(**L**) Quantification of ribbon bodies per IHCs and OHCs in P8. All differences were not significant (ns). Sample sizes: n = 6 for the control group, which is a combination of heterozygous and WT littermates; 4 *Sema3f−/−*. The scale bar in **B** = 15 μm for **A** and **B**; 30 μm, **D**; 40 μm, **F** and 12 μm, **H**–**K**.

## Plexin-A3 is expressed by the SGNs during development

It was shown previously that intracellular-signaling events that occur as a result of Sema3f/Nrp2 interactions depend entirely on the presence of Plexin-A3 (*Yaron et al., 2005*). Therefore, we examined Plexin-A3 expression in the SGNs during developmental periods corresponding with IHC and OHC innervation (*Figure 9—figure supplement 1*, *Figure 9*). In situ hybridization data from the Allen Brain Atlas Website clearly show *Plxna3* mRNA distribution in the cochleovestibular ganglion at E11.5 and in the SGNs at E13.5 and E15.5 (http://developingmouse.brain-map.org/gene/show/18610) (*Figure 9—figure supplement 1*). To verify these results, we generated *Plxna3* sense and antisense mRNA probes and performed in situ hybridization at E16.5. Consistent with the Allen Brain Atlas, *Plxna3* antisense probes indicated broad expression in SGNs in both the apical and basal regions of the cochlea (*Figure 9A–C*). Using differential interference contrast to closely examine the SGNs (*Figure 9D–F*), we did not detect any obvious differences in *Plxna3* mRNA within the population, suggesting uniform expression of *Plxna3*. In contrast, examination of Plexin-A3 protein distribution in whole-mount preparations at E16.5 indicated a striking difference in Plexin-A3 levels in SGN peripheral processes. At low magnification, Plexin-A3 protein is visible broadly in the SGN processes in a pattern similar to that of Tuj1 (*Figure 9G–I*). However, at high-magnification, Plexin-A3 expression in fibers extending beyond the IHC region is markedly reduced by comparison with substantially higher levels in fibers that terminate near IHCs (*Figure 9G–L*). Since SGNs are actively probing the IHC and OHC regions at this stage (*Figures 1, 2*), individual fibers cannot unambiguously be designated as 'type I' or 'type II'. But, Plexin-A3 expression on fibers located in the OHC region is detectable only on sparse numbers of SGNs as small puncta (arrowheads). These data suggest that Plexin-A3 may act as an Nrp2 co-receptor in SGNs, and that differences in Plexin-A3 protein levels could modulate Sema3F-mediated repulsion.

## Discussion

The SGN comprises multiple neuronal phenotypes arranged into precise patterns of innervation. Despite initial descriptions dating back more than 100 years, our understanding of the development and organization of this structure remains limited. A particularly intriguing issue has been the question of whether type I and type II SGNs are determined as a result of the expression of distinct transcriptional networks, as a result of interactions between peripheral axons and target HCs, or a combination of both. Classic studies of developing SGNs by Echteler and others indicated the presence of individual fibers in contact with both IHCs and OHCs, supporting the possibility of target-mediated determination of SGN cell types (*Echteler, 1992*). However, more recent work using genetic labeling provides strong evidence for the presence of committed type I and type II SGN fibers in the mouse inner ear even at embryonic ages (*Koundakjian et al., 2007*). Moreover, detailed morphological and cellular analyses by N Druckenbroad and L Goodrich (personal communication) revealed that both type I and II SGNs show stereotyped path-finding behaviors, supporting the idea that each SGN subtype may be predetermined to an extent. The results presented here (and by Druckenbroad and Goodrich) show that cochlear innervation normally involves the transient extension of SGN fibers into the OHC region between E15.5 and E16.5. After this brief period and continuing through P0, the majority of SGN processes converge on the IHC region. This occurs through both the arrival of new fibers and the retraction of existing processes from the OHC region. Labeling of future type I fibers with transient OHC extensions could appear as individual neurons with indeterminate phenotypes. Moreover, considering that the cells within the cochlear epithelium are in the process of both differentiation and active extension between E15.5 and E16.5 (*Yamamoto et al., 2009*; *Wu and Kelley, 2012*; *Coate and Kelley, 2013*), it is not surprising that SGN processes might also be highly dynamic during this time period. Nrp2-mediated repulsion in response to Sema3F was identified many years ago and has been documented in multiple events ranging from axon guidance to synaptic pruning (*Adams et al., 1997*; *Chen et al., 1997*; *Giger et al., 2000*; *Demyanenko et al., 2014*). Therefore, there is strong precedent for Sema3F-Nrp2 interactions serving a repulsive role in SGN-HC connectivity.

The observation of active extension and retraction of SGN processes into and out of the OHC region is consistent with the presence of inhibitory signals acting on those processes. In these studies, we demonstrate that repulsive signaling mediated by Sema3F in the OHC region and Nrp2 in SGNs controls, at least in part, the number of SGN fibers that terminate on OHCs. Our findings support a model in which the embryonic OHC region maintains a compartment of Sema3F protein expression, which activates Nrp2 receptors expressed on SGNs leading to retraction of processes (*Figure 8D*). Since this inhibitory signal is restricted to the OHC domain, this interaction would enforce contacts between SGN fibers and IHCs.

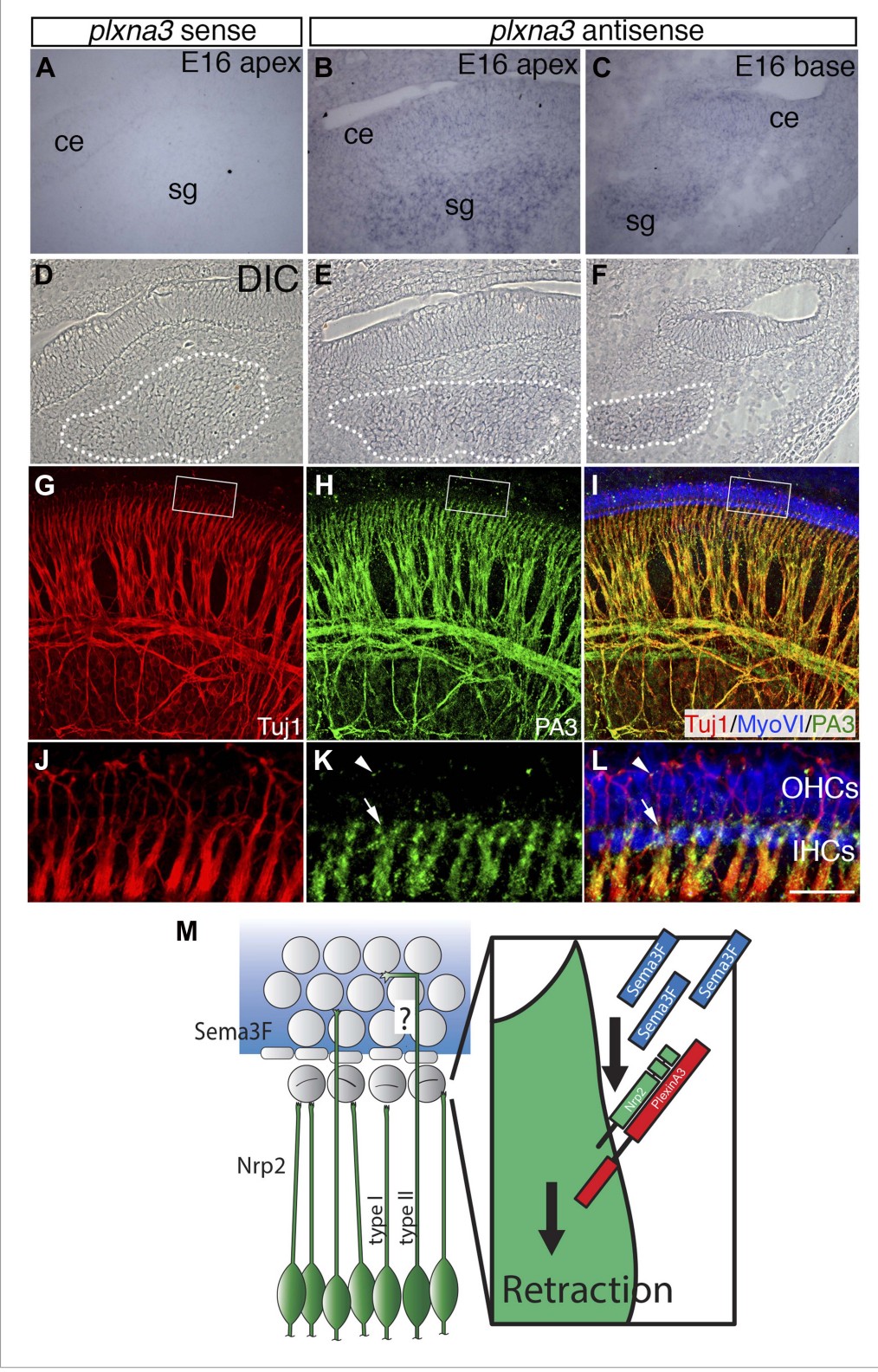

**Figure 9**. Differential expression of Plexin-A3 in SGNs and a proposed model. (**A–F**) In situ hybridization for *Plxna3* at E16.5. (**A**) A *Plxna3* sense control probe does not show any reactivity. (**B**, **C**) *Plxna3* transcripts are present in the SG, and possibly faintly in cells throughout the cochlear epithelium (ce). (**D–F**) Differential interference contrast (DIC) images for **A–C** were used to identify SGNs by morphology. The white dotted lines encircle the SGNs in each panel. *Figure 9. continued on next page*

*Figure 9. Continued*

(**G–I**) Plexin-A3 antibody staining in an E16.5 whole-mount cochlea. (**G–I**) Expression of Plexin-A3 (green) in SGNs appears to overlap with a neuronal marker (Tuj1 in red) except in the HC region (MyoVI staining in blue). (**J–L**) High-magnification views from the boxed regions in **G**, **H**, and **I**. Plexin-A3 is strongly expressed in SGN peripheral processes located adjacent to the IHC region (arrows) but appears markedly reduced in fibers that cross into the OHC region. Limited Plexin-A3 is detectable on sparse numbers of SGN processes in the OHC region (arrowhead). Scale bar in **L**: 100 µm, **A–I**, 25 µm, **J–L**. (**M**) Proposed model: Sema3F (blue shading and blue rectangles) is expressed by cells in the OHC region of the cochlea with a sharp boundary along the PCs. Binding of Sema3F to Nrp2 (green SGNs and green rectangle) induces retraction events that inhibit the growth of the type I SGN peripheral processes into the OHC domain. PlexinA3 co-receptors, which likely serve a role in this process, are shown in red. '?' illustrates an outstanding question of how type II SGNs, which express Nrp2 and are confronted by Sema3F, are able to extend into the OHC domain.

The following figure supplement is available for figure 9:

**Figure supplement 1**. *Plxna3* expression in the inner ear at E11.5-E15.5.

Type I SGNs were still observed even in the complete absence of either *Nrp2* or *Sema3f*. There are several possible explanations for this phenotype. First, it is reasonable to assume that compensatory guidance mechanisms are in place to ensure that most type I SGNs terminate on IHCs. For example, there are a variety of cell adhesion molecules (CAMs) expressed in the cochlea such as the immunoglobulin superfamily of CAMs (like NCAM), cadherins (E-cad and N-Cad), and integrins (*Kelley, 2003*), all of which may help attract type I SGNs to the IHC region. In addition, it was recently shown that ephrin-A5/Epha4 signaling may also act to repel type I SGNs from OHCs (*Defourny et al., 2013*). These studies, however, were limited to early postnatal and adult cochlear samples, thus, the extent to which these factors are involved in targeting events during embryonic development remains uncertain. Finally, it is also possible that the developing SGN population is heterogeneous with some neurons demonstrating more dynamic exploratory behavior than others. Therefore, some fibers may be less inclined to extend into the OHC region even in the absence of inhibitory signals. This hypothesis might be consistent with the idea that some SGNs are initially committed to a type I fate while others rely on target-based interactions to determine their phenotype.

As discussed, despite the presence of additional type II-like processes in $Nrp2^{+/-}$ cochleae at birth, analysis of adult animals indicated a normal number of type II SGNs and hearing sensitivity. These results suggest that the additional type II-like fibers observed at birth are most likely eliminated over time and the factors that mediate this recovery are unknown. Given that 'mature' type I/type II SGN patterns are established by P0 (*Figure 2*) and that HC ribbon synapses are visible by P0 (*Huang et al., 2012*), it could be that Eph/ephrin signaling might be involved in synaptic pruning-based refinements postnatally. In addition, programmed cell death may also contribute to the final innervation pattern, as apoptosis in SGNs has been reported in the postnatal gerbil (*Echteler et al., 2005*). However, we did not detect any appreciable SGN fragmentation or apoptosis in our experiments here (*Figure 2—figure supplement 1*) or in previous studies (*Coate et al., 2012*). Therefore, while we cannot rule it out entirely, we would not predict that apoptosis plays a significant role in shaping cochlear innervation during development.

Finally, this study also raises several intriguing questions to be addressed in the future. First, since Sema3F acts as a repulsive signal for SGN processes, how are type II SGNs, which appear to express Nrp2 (*Figure 2*), able to project into the OHC region (see '?' in *Figure 9*)? We did not detect differences in Nrp2 levels between putative type I and type II SGNs, which could have accounted possibly for differences in responsiveness to Sema3F. We are currently in favor of the model that type I and type II SGNs differentially express or target Plexin-A co-receptors, as the absence of these obligatory co-receptors could effectively inactive the Sema3F-Nrp2 inhibitory interaction. Plexin-A1 and Plexin-A3 proteins were localized to the SGNs previously (*Murakami et al., 2001*; *Katayama et al., 2013*). Here, we found *Plxna3* mRNA broadly expressed by the SGNs and Plexin-A3 protein enriched in putative type I SGNs (*Figure 9*). While these data suggest that where type I or II SGNs ultimately terminate may depend on Plexin-A3 protein levels, a more careful analysis of the Plexin-A3 distribution and function is needed. A second question relates to the remarkably different patterns of expression exhibited by the different *Sema3s* (*Figure 6*). What factor(s) control these discrete *Sema3* domains?

GATA transcription factors have been shown previously to regulate *Sema* expression and are known players in inner ear morphogenesis and wiring (*Lepore et al., 2006*; *Kodo et al., 2009*; *Appler and Lu, 2013*), so perhaps one role of GATAs in the inner ear is to regulate *Sema3* levels. Other clues related to this matter may come from previous findings where either the loss of Prox1 or Fgfr3 signaling led to increases in OHC innervation (*Puligilla et al., 2007*; *Fritzsch et al., 2010*). It is possible, therefore, that Prox1 controls the expression of factors such as *Sema3f* or that the local distribution of the different *Sema3s* is controlled by morphogens such as Fgfs. Overall, it will be important in future studies to investigate upstream activators of *Sema3* expression and mechanisms that ultimately distinguish the morphological differentiation of type I and type II SGNs.

## Materials and methods

### Mouse lines and tissue preparation

All animals used in this study were maintained in accordance with the NIH Animal Care and Use Committee. For expression studies and in vitro culture experiments, timed-pregnant CD1 mice (Charles River Laboratories, Frederick, MD) were used. The *Neurog1^{CreERT2}* line used to characterize SGN phenotypes has been described previously (*Koundakjian et al., 2007*). These mice were crossed with a *Rosa26^{tdTomato}* reporter line (Jackson Laboratories, Bar Harbor, ME; stock# 007914) and with an *Atoh1^{nGFP}* line (*Lumpkin et al., 2003*) to generate triple transgenics referred to as 'N^{Cre};R^{Tom}; *Atoh1^{nGFP}*'. Because a small amount of spontaneous Cre activity within the *Neurog1^{CreERT2}* line resulted in recombination and expression of tdTomato in a limited number of SGNs, no tamoxifen was administered to the dams. No apparent regional or cell type bias in the pattern of spontaneous recombination was observed along the tonotopic (basal-to-apical) axis or between type I and type II SGNs. The *Nrp2* and *Sema3f* mutant lines have been described previously (*Giger et al., 2000*; *Walz et al., 2007*).

For immunohistochemical studies of whole mounts, cochleae were removed from the skull and fixed in 4% paraformaldehyde (PFA; 30 min to 1 hr depending on age) at RT followed by extensive rinsing in cold 1× phosphate-buffered saline (PBS). For tissue sectioning, following fixation, inner ears were exposed to increasing concentrations of sucrose (up to 30%) and then embedded and frozen in optimal cutting temperature (OCT) medium (Sakura Finetek, Torrance, CA). In most cases, OCT blocks were sectioned at 12 µm. In situ hybridizations were performed on frozen sections that had been prepared similarly, but with overnight fixation prior to embedding.

### Antibodies, immunohistochemistry, and in situ hybridization

The antibodies and concentrations used in this study were as follows: mouse-anti-Tuj1 (Covance, Chantilly, CA); 1:500, rabbit-anti-dsRed (Clontech, Mountain View, CA); 1:500, chicken-anti-Atoh1 (*Driver et al., 2013*); 1:10,000, chicken-anti-NFS (Aves Labs, Tigard, OR); 1:1,000, rabbit-anti-myosin VI (Proteus Biosciences, Ramona, CA); 1:1,000, goat-anti-Sox2 (Santa Cruz Biotechnology, Dallas, TX) rabbit-anti-Nrp2 for sections (*Giger et al., 1998*); 1:500, goat-anti-Nrp2 for whole-mount staining; 1:500 (R&D Systems, Minneapolis MN), mouse-anti-human Fc; 1:100 (Jackson Immunoresearch, West Grove, PA), chicken-anti-synaptotagmin 1 (Aves Labs); 1:1,000, mouse-anti-GAP43 (Chemicon, Billerica, MA); 1:1,000, chicken-anti-GFP (Aves Labs); 1:1,000, mouse-anti-CtBp2 (BD Biosciences, San Jose, CA); 1:200, rabbit-anti-cleaved caspase 3; 1:500, and rabbit-anti-CDKN1B (formerly p27^{Kip1}; Novus Biologicals, Littleton, CO). A goat polyclonal antibody against myosin VI was generated (Pacific Immunology, Ramona, CA) using an immunizing fusion protein provided by Proteus Biosciences (*Hasson and Mooseker, 1994*); 1:1000. To detect Plexin-A3, we used a goat-anti-Plexin-A3 antibody from R&D Systems (catalog# AF4075). This antibody was raised against the extracellular domain of mouse and rat Plexin-A3 (Leu34-Asp150) and, according to the manufacturer, shows less than 5% cross-reactivity with other Plexins. This antibody was validated by R&D Systems using ELISA, Western blot, and immunofluorescence. The Plexin-A3 protein staining we report here (in SGNs) is consistent with the *Plxna3* mRNA distribution also reported here and noted previously (*Murakami et al., 2001*). Actin was detected using phalloidin conjugates (Life Technologies, Carlsbad, CA) at 1:100.

After fixation, whole-cochleae or tissue sections were permeabilized with PBS supplemented with 0.5% Triton-X100 (Sigma, St. Louis, MO) for 20 min at RT. Tissue samples were then exposed to either 10% normal goat or horse serum (Vector Labs, Burlingame, CA) for 1 hr. When anti-mouse secondary antibodies were used, anti-mouse FAB fragments (Jackson Immunoresearch) were added to the blocking solution at 1:200. When anti-chicken secondary antibodies were used, 10% Blokhen Reagent

(Aves Labs) was added. Following blocking, samples were incubated in primary antibody (diluted in PBS +0.5% Triton) overnight at 4°C. After extensive rinsing the following day, 488-, 555-, or 633-Alexa Fluor secondary antibodies (Life Technologies) were used for detection of primary antibody binding. Images were acquired using a Zeiss LSM-510 or -710 laser scanning confocal microscope and then further processed using ImageJ and/or Adobe Photoshop software. For the experiments looking at Nrp2 expression over developmental time by tissue sectioning in *Figure 3*, identical processing, immunostaining and image acquisition procedures and settings were used.

In situ hybridization was performed as described (*Driver et al., 2008*), but with antisense probes generated from mouse, *Sema3f* (NM_011349), *Sema3b* (BC150758), *Sema3c* (BC066852.1), *Sema3d* (BC138131.1), and *Plxna3* (NM_008883) templates. For every experiment, sense probes were generated and used as negative controls. The images representing *Plxna3* from the Allen Brain Atlas database can be found here: http://developingmouse.brain-map.org/gene/show/18610.

### Binding assays with Fc-conjugated probes

To localize endogenous binding partners for Sema3F, we used Sema3F-Fc-conjugated protein as previously described (*Sahay et al., 2003*), but with minor modifications. Briefly, whole-inner ears were fixed briefly in ice-cold 4% paraformaldehyde to help maintain structural stability and processed for cryosectioning. Without permeabilizing, 12-μm sections were rinsed in 1× PBS then incubated with 20 nM human Fc (Jackson Immunoresearch) or Sema3F-Fc (R&D Systems) overnight at 4°C. After extensive rinsing with PBS, the tissue samples were treated with 4% PFA for 15 min, rinsed extensively in 1× PBS, then processed for anti-Fc and anti-NF200 immunostaining.

### Western blotting

P0 cochleae were isolated and digested in radio immunoprecipitation assay (RIPA) buffer (Sigma) supplemented with protease inhibitors (Roche, Basel, Switzerland). Lysates were run on a 4–12% acrylamide gel and then transferred to nitrocellulose membranes. Membranes were then blocked with 5% milk block (Blotto; Santa Cruz Biotechnology, Dallas, TX) in tris-buffered saline with tween (TBST) and then probed with either rabbit-anti-Nrp2 (1:2000; R&D Systems; *Demyanenko et al., 2011*), or chicken-ant-Tuj1 antibodies (1:5000, Aves) overnight. The following day, following extensive rinsing, the membranes were probed with peroxidase-conjugated secondary antibodies (1:5000; Jackson Immunoresearch) for 45 min, treated with Clarity chemiluminescence substrates (Bio-Rad, Hercules, CA), and imaged using an Imagequant LAS 4000 (GE Life Sciences, Pittsburgh, PA).

### Live tissue imaging

N$^{Cre}$;R$^{Tom}$; *Atoh1$^{nGFP}$*-positive embryos or postnatal pups were identified using a dissection microscope with fluorescence illumination. Heads were place in chilled 1× Hank's buffered saline solution (HBSS), inner ears were dissected, and the cochlear capsule, stria vascularis, and Reissner's membrane were carefully removed. Each semi-intact cochlea was then incubated on a polycarbonate membrane filter (Sterlitech, Kent, WA) for 2 hr at 37°C to gently flatten the tissue prior to imaging. After this 2-hr period, each cochlea was transferred to a Mattek dish (Mattek Corporation, Ashland, MA) and flattened next to the cover glass with a platinum harp strung with nylon filament. The imaging medium included Leibovitz medium (Life Technologies), 10% fetal bovine serum, 0.2% N2, 0.001% Ciprofloxacin, and 0.1 mM Trolox (as an antioxidant). Using an Axiovision (Zeiss, Oberkochen, Germany) spinning disk confocal microscope, the SGNs and HCs were imaged at a capture speed of 250–400 ms at 10-to 20-min intervals. Time-lapse data were subsequently processed using Volocity software (Perkin Elmer, Waltham, MA).

### Cochlear culture experiments

For experiments using Fc fusion proteins, E15.5 or E16.5 cochleae from N$^{Cre}$;R$^{Tom}$ embryos were prepared as described above, but maintained on polycarbonate filters (Sterlitech) for 24 hr. In these experiments, the culture medium contained Dulbecco's Modified Eagle Medium (DMEM), 10% fetal bovine serum, 0.2% N2, and 0.001% ciprofloxacin. Purified human IgG-Fc (Jackson Immunoresearch) or Sema3F-Fc (R&D Systems) was added directly to the medium to a final concentration of 20 nM. Following the culture period, the cochleae were fixed for 15 min in chilled 4% PFA, then rinsed extensively before immunostaining and confocal imaging.

## Quantification methods

To quantify the developmental time series for type I and type II projection patterns in *Figure 2*, confocal Z-stacks were acquired from the base of each cochlea at positions corresponding to 10–25% of the cochlear length. An SGN was considered 'in the OHC region' if the most prominent part of the axon (barrel) clearly crossed the plane of the medial edge of the first OHC. Conversely, an SGN was considered 'in the IHC region' if it stopped at the IHC region. To quantify the number of Syt+ axons that had crossed into the OHC region in *Figure 3*, confocal Z-stacks were acquired at positions corresponding to 50% of the cochlear length. The number of Syt+ fibers that crossed into the OHC region were counted and normalized to the length of the region; the length of the region was determined by drawing a line along the medial side of the IHCs. For quantifications using the sparse SGN-labeling model (*Figures 4, 8*), a series of high-resolution confocal Z-stacks were acquired along the length of each specimen using an automated-scanning stage. The files were subsequently imported to Volocity for quantification of IHC vs OHC position and branching morphometrics. To quantify ribbon synapses, cochleae were stained with anti-CtBP2 antibodies and photographed as described previously (*Yu et al., 2013*). To quantify E15.5 SGN density in *Figure 3*, Tuj1-stained whole-mount cochleae were imaged by confocal z-stack and the spherical nuclei within each stack were counted manually. For the SGN counts in *Figure 8*, similar methods were used, but we were only able to reliably quantify the first few cell layers because of limited tissue penetration by the mouse-anti-Tuj1 antibody at this older stage (P0). For consistency across the P0 samples, we limited our counts to only 10 µm of z depth. To quantify HC density in *Figure 8*, myosin VI-positive HCs at the base of P0 cochleae were imaged by confocal microscopy. For each sample, we acquired 5 confocal z-stacks at 63× in non-overlapping regions along the first 1 mm of cochlear length (starting from the base). HCs were counted manually.

To quantify apoptosis in E16.5 and E17.5 SGNs, we stained cochleae in whole mount with anti-cleaved caspase-3 (CC3) and Tuj1 antibodies. Cells were counted as SGNs undergoing apoptosis if they were (1) CC3 positive, (2) within the Tuj1-positive SGN cell body region, and (3) bipolar cells (7–8 µm wide) with peripheral and central projections. We could not rely on Tuj1 co-staining because the Tuj1 antibody did not show identical levels of tissue penetration compared to the CC3 antibodies. Other cells (glial cells, blood vessel cells, mesenchyme cells) or fragmented debris were not counted. Low-magnification views were used to measure cochlear length in order to determine the basal and apical halves. Two-tailed *t*-tests were used throughout. Sample numbers are indicated in the figure legends.

## Acknowledgements

We thank the members of the Kelley and Coate laboratories for their valuable discussions and technical assistance during this work. We also thank Drs Lisa Goodrich and Noah Drukenbrod (Harvard University) for their very insightful discussions. We thank Dr Wei-Ming Yu (Harvard University) for insight on ribbon synapse staining and Dr Kenneth Yamada (NIH/NICHD) for his helpful assistance with time-lapse imaging. We also thank Mr Jacob Salzberg for his assistance with morphometric quantifications. We thank Drs Jeffrey Huang (Georgetown University) and Katherine Kindt (NIDCD) for the critical reading of this manuscript. The *Nrp2* null mouse and the anti-Nrp2 antibodies were kind gifts from Dr Alex Kolodkin, Johns Hopkins University. The *Neurog1^{CreERT2}* mouse line was a kind gift from Dr Lisa Goodrich, Harvard University. The National Institute on Deafness and Other Communication Disorders funded this work through the Intramural Research Program (to MWK, DC000059) and DC13107 (to TMC).

## Additional information

### Funding

| Funder | Grant reference | Author |
| --- | --- | --- |
| National Institute on Deafness and Other Communication Disorders (NIDCD) | DC13107 | Thomas M Coate |

| Funder | Grant reference | Author |
| --- | --- | --- |
| National Institute on Deafness and Other Communication Disorders (NIDCD) | DC000059 | Matthew W Kelley |

The funder had no role in study design, data collection and interpretation, or the decision to submit the work for publication.

### Author contributions

TMC, Conception and design, Acquisition of data, Analysis and interpretation of data, Drafting or revising the article; NAS, KTI, Acquisition of data, Analysis and interpretation of data; KDZ, Acquisition of data, Analysis and interpretation of data, Drafting or revising the article; MWK, Conception and design, Analysis and interpretation of data, Drafting or revising the article

### Ethics

Animal experimentation: All animals used in this study were maintained in accordance with the NIH Animal Care and Use Committee, Protocol #1262, and the Georgetown University Animal Care and Use Committee, Protocol # 14-040-100179.

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
