## [Decision Letter]

Thank you for submitting your work entitled “Neuropilin-2/Semaphorin-3F-mediated Repulsion Promotes Inner Hair Cell Innervation by Spiral Ganglion Neurons” for peer review at *eLife*. Your submission has been favorably evaluated by a Senior editor, a Reviewing editor, and three reviewers.

The reviewers have discussed the reviews with one another and the Reviewing editor has drafted this decision to help you prepare a revised submission.

Summary:

This work investigates the selective innervation by type I spinal ganglion neurons (SGNs) of inner hair cells in the developing murine cochlea, demonstrating selective retraction of SGNs from outer hair cells prior to the likely formation of synapses with inner hair cells (IHCs). In addition to observing SGN innervation of both IHCs and OHCs, these experiments suggest that SGNs ultimately innervating IHCs appear to prune back contacts with OHCs. Determination of the number of SGCs innervating OHCs and IHCs, coupled with time-lapse imaging of SGN innervation, provide support this model. To understand the molecular basis for selective SGN CE innervation, the authors address the significance of neuropilin-2 (Nrp2) expression they observe in SGN neurons, apparently in all type 1 and II SGNs, during the developmental time window when projections are segregated. In Nrp2 mutants (both -/- and +/-) a general increase in CE innervation in the OHC region is observed, and using Syt-1 as an SGN-specific marker this hyper-innervation is shown to be due to SGN axons. More detailed genetic labeling in Nrp2-/+ mutants leads to a similar conclusion, and it also allows for observations of increased filopodial extensions from SGNs in the IHC CE. Surprisingly, both ribbon synapse number in cochleae (OHC and IHC), and the overall percentage of type II SGNs as determined morphologically by genetic labeling, are unchanged in adult Nrp2-/+ animals. Sema3F expression by in situ analysis correlates very nicely with this cue playing a role in constraining type I SGN extension to the IHC region. In vitro cochlea organotypic cultures are used to show that treatment with exogenous Sema3F can affect the extension and branching of SGN neurons, consistent with the proposed role for Sema3F in SGN connectivity. Finally, assessment using sparse genetic labeling of SGN innervation in a Sema3F null mutant shows photocopies the SGN targeting defects observed in Nrp2-/+ mutants.

There are, however, weaknesses that should be straightforward to address and doing so would significantly strengthen support for the proposed models. These weaknesses include reliance on the Nrp2/+animals, limited analyses of the role of Sema3F and a need to evaluate the role of A-type plexins, which could be differentially expressed on type I and II SGN axons. Specific comments are detailed below:

Major criticisms:

1) The apparent expression of Nrp2 in all SGN neurons presents a challenge to understanding the selective guidance of Type I SGN afferents by Sema3F as described here. The Nrp2 IHC presented in Figure 1 is not as convincing with respect to process distribution as it might be, and it would be helpful to complement these data with Alkaline-phosphatase::Sema3F ligand-binding to similar preparations at multiple states of cochlear development. This is a very sensitive assay, and it would indicate whether there is distinct subcellular localization of Nrp2 in type 2 as compared to type 1 SGN axons. If this is not possible, or lacks sufficient resolution in the hands of the authors, then analysis in flat mount tissue at higher magnification should be performed.

2) Neuropilin-2 staining should also be done on whole mount cochlea (not only sections) and on Ngn1-CreERT2;Rosatomato. This point is very important as a major problem with the proposed model is the presence of neuropilin-2 on both type I and type II SGN axons.

3) The authors should assess PlexA3 mRNA expression in SGN cell bodies throughout cochlear development since this is another potential determinant of Sema3F-selective guidance of SGN afferents.

4) The phenotypic analysis of the Sema3f KO is a bit too short (the age is mentioned neither in the legend nor on the figure). The authors should determine:

A) Is the defect is also transient in the Sema3f KO ?

B) Is the number of SGN neurons or IHCs/OHCs normal in the KO?

C) Do axons that remain in the OHCs region express neuropilin-2?

D) Can the authors clarify if there is any cell death that occurs in SGN neurons during the normal developmental intervals under investigation here?

E) Is SGN cell number affected at all in Nrp2-/-, Nrp2+/- and, importantly, in Sema3F-/- mutants?

5) Throughout this manuscript the authors refer to type I and II neuronal morphologies observed using their sparse labeling approach in terms of “numbers” of these two neuronal classes. However, the figures and methods make clear that this is not an absolute determination of cell number, but the % of sparsely labeled cells with these morphological characteristics that is presented. This should be changed throughout the text for experimental clarity.

---

## [Author Response]

*1) The apparent expression of Nrp2 in all SGN neurons presents a challenge to understanding the selective guidance of Type I SGN afferents by Sema3F as described here. The Nrp2 IHC presented in*
Figure 1
*is not as convincing with respect to process distribution as it might be, and it would be helpful to complement these data with Alkaline-phosphatase::Sema3F ligand-binding to similar preparations at multiple states of cochlear development. This is a very sensitive assay, and it would indicate whether there is distinct subcellular localization of Nrp2 in type 2 as compared to type 1 SGN axons. If this is not possible, or lacks sufficient resolution in the hands of the authors, then analysis in flat mount tissue at higher magnification should be performed*.

We appreciate this very insightful idea and have performed the experiments, but with minor modification. We were concerned that the colorimetric reaction used to detect AP-tagged fusion proteins would not provide the optical resolution necessary to identify different SGN or hair cell subpopulations. As an alternative, we used Fc-tagged fusion proteins and detected them by anti-Fc immunostaining and confocal microscopy. We were able to detect Sema-3F-Fc on both type I and II SGNs, similar to what was evident by the anti-Nrp2 antibody staining. The data appears now as Figure 3—figure supplement 1. Text additions for these experiments can be found in the Results section (subsection “Nrp2 is expressed in SGNs and is required for a normal pattern of innervation”), the Materials and Methods (subsection “Binding assays with Fc conjugated probes”), and there is a new figure legend that can be found at Figure 2—figure supplement 1. The results support our original conclusion, that Nrp2 is expressed on both type I and type II SGNs.

However, as is described below in response to Major Criticism 3, we demonstrate that Plexin-A3 is differentially expressed between type I and type II SGNs. Therefore, it seems likely that it is the lack of Plexin-A3, rather than Nrp2, that mediates the differential responses of type I and type II SGNs to Sema-3F.

2) Neuropilin-2 staining should also be done on whole mount cochlea (not only sections) and on Ngn1-CreERT2;Rosatomato. This point is very important as a major problem with the proposed model is the presence of neuropilin-2 on both type I and type II SGN axons.

This was completed and appears now in Figure 3. In this new version, we have removed the P0 cross-section that appeared in the original figure and replaced it with the micrograph illustrating the SGN peripheral processes in whole mount. The original P0 sample was used to document our observation that Nrp2 levels in the SGN cell bodies begin to diminish around the time of birth. We have left this note in the text. Nrp2 immunoreactivity is still clearly evident in the SGN processes at this time point, however. P0 is the ideal time to show this, as type I and type II SGNs have been fully segregated by this stage (as documented in Figures 1 and 2). During the time frame for submission of revisions, we were not able to collect enough *Neurog1*^*CreERT2*^*; tdTomato*-positive tissue to complement the whole mount preparations done with WT tissue. However, in our view, the Nrp2 staining in WT samples is entirely consistent with the position of type II SGNs and the results presented in response to Criticism 1. Text additions for this experiment can be found in the Results section (subsection “Nrp2 is expressed in SGNs and is required for a normal pattern of innervation”), in the Materials and methods (subsection “Antibodies, immunohistochemistry and in situ hybridization”) and changes were made to the existing figure legend.

*3) The authors should assess PlexA3 mRNA expression in SGN cell bodies throughout cochlear development since this is another potential determinant of Sema3F-selective guidance of SGN afferents*.

*Plxna3* mRNA expression data for several embryonic stages of mouse development is already available on the Allen Brain Atlas website. *Plxna3* mRNA is clearly visible in neurons of the cochleovestibular ganglion (SGN precursors) at E11.5 and in SGNs at E13.5 and E15.5. It is the generous policy of the Allen Brain Atlas that data from their website can be used and displayed in publications provided appropriate citations are made. The policy can be viewed here: http://www.alleninstitute.org/citation-policy/. Therefore, we have elected to cite these data and display example images as Figure 9—figure supplement 1. In addition, to confirm their results, we generated our own *Plxna3*-specific riboprobe and found that it labels all of the SGN cell bodies at E16.5. These data were added to Figure 9. In addition, we also show Plexin-A3 antibody staining. The antibody staining is very intriguing, as Plexin-A3 is apparently not expressed on type II SGNs. Therefore, it may be the distribution of Plexin-A3 that accounts for the selective guidance of type I and type II SGNs. These data are described in the Results section (subsection “Plexin-A3 is expressed by the SGNs during development”), the Materials and methods section (subsection “Antibodies, immunohistochemistry and in situ hybridization”), in the Figure 9 legend and in the Discussion (fifth paragraph).

*4) The phenotypic analysis of the Sema3f KO is a bit too short (the age is mentioned neither in the legend nor on the figure)*.

Thank you for pointing out that we failed to include the age (P0). This was added to the figure and legend.

*The authors should determine*:

A) Is the defect is also transient in the Sema3f KO ?

Yes, as shown in new data in Figure 8, we found the defect is also transient in the *Sema3f ko* mice. Because of the time period provided for revisions and several cannibalized litters, we were only able to generate 4 *Sema3f ko* cochleae and 6 controls. These 6 controls consisted of 2 WT cochleae and 4 *Sema3f* heterozygous cochleae. No differences were observed between *Sema3f hets* and WT, so these animals were pooled as controls. There were no significant differences between the mutant and control cells and as is illustrated in Figure 8, standard deviations were very small, suggesting that the results are consistent despite the limited sample size. These data are described in the Results section (subsection “Deletion of Sema3f leads to increased SGNs in the OHC region”) and in the Figure 8 legend.

B) Is the number of SGN neurons or IHCs/OHCs normal in the KO?

Yes, SGN and IHC/OHC numbers are normal in the KO. We have added these results to Figure 8. Text associated with these experiments can be found in the Results section (subsection “Deletion of Sema3fleads to increased SGNs in the OHC region”), the Materials and methods section (subsection “Quantification methods”), and to the Figure 8 legend.

C) Do axons that remain in the OHCs region express neuropilin-2?

This is an excellent question, but one that is unfortunately impossible for us to address. The tdTomato reporter labels all SGNs uniformly; therefore, we do not have any way to determine which SGNs in the OHC region are extra remaining SGNs vs. those that would constitute the normal population.

D) Can the authors clarify if there is any cell death that occurs in SGN neurons during the normal developmental intervals under investigation here?

We appreciate this request, as we agree that clarifying this issue helps strengthen the manuscript. We had previously examined apoptosis in cochleae by cross-section ([9], Neuron), but with a focus on otic mesenchyme cells, not neurons. So, we elected here to repeat these studies using whole-mount E16.5 and E17.5 cochleae and with a focus on the SGNs. These data can be found as Figure 2—figure supplement 1. Overall, there were very few apoptotic neurons observed, thus it seems unlikely that neuronal cell death plays a significant role in SGN development at these stages. In the text, this is addressed in the Results section (subsection “Maturing SGNs extend and then retract projections from the OHC region”), the Materials and methods section (subsection “Quantification methods”) and in the Figure 2—figure supplement 1 legend. We also included some brief changes to the Discussion section (fourth paragraph).

*E) Is SGN cell number affected at all in Nrp2-/-, Nrp2+/- and, importantly*, *in Sema3F-/- mutants?*

Data for SGN cell numbers (as density measurements) in WT and *Sema3f*^*-/-*^ cochleae has been added to Figure 8. No changes in SGN density were observed in *Sema3f* mutants. These new data are described in the Results section (subsection “Deletion of Sema3f leads to increased SGNs in the OHC region”) in the Materials and methods (subsection “Quantification methods”) and in the legend for Figure 8.

Analysis of SGN cell number for the *Nrp2* mutant mice appeared in the original submission in Figure 3 and is described in the Results section. We apologize for not making this more conspicuous. For better clarity, we altered the y axis label on Figure 3 to read “SGN somata density.” We did not have suitable numbers of E15.5 *Nrp2*^*+/-*^ cochleae to add to the data so we have left this data in the form that was originally submitted. However, given that there are no differences between WT and *Nrp2*^*-/-*^ cochleae in terms of SGNs numbers, it seems very unlikely that the results for *Nrp2*^*+/-*^ cochleae would differ from either of these two groups.

*5) Throughout this manuscript the authors refer to type I and II neuronal morphologies observed using their sparse labeling approach in terms of “numbers” of these two neuronal classes. However, the figures and methods make clear that this is not an absolute determination of cell number, but the % of sparsely labeled cells with these morphological characteristics that is presented. This should be changed throughout the text for experimental clarity*.

Thank you for pointing this out. We have now attempted to add as much clarity to the language in the text and made sure the term “percentage” was used when appropriate.